# Data-driven prediction of colonization outcomes for complex microbial communities

Lu Wu [1,6], Xu-Wen Wang [2,6], Zining Tao[1,3], Tong Wang [2], Wenlong Zuo [1], Yu Zeng[1], Yang-Yu Liu [2,4] ✉ & Lei Dai [1,5] ✉

Microbial interactions can lead to different colonization outcomes of exogenous species, be they pathogenic or beneficial in nature. Predicting the colonization of exogenous species in complex communities remains a fundamental challenge in microbial ecology, mainly due to our limited knowledge of the diverse mechanisms governing microbial dynamics. Here, we propose a data-driven approach independent of any dynamics model to predict colonization outcomes of exogenous species from the baseline compositions of microbial communities. We systematically validate this approach using synthetic data, finding that machine learning models can predict not only the binary colonization outcome but also the post-invasion steady-state abundance of the invading species. Then we conduct colonization experiments for commensal gut bacteria species *Enterococcus faecium* and *Akkermansia muciniphila* in hundreds of human stool-derived in vitro microbial communities, confirming that the data-driven approaches can predict the colonization outcomes in experiments. Furthermore, we find that while most resident species are predicted to have a weak negative impact on the colonization of exogenous species, strongly interacting species could significantly alter the colonization outcomes, e.g., *Enterococcus faecalis* inhibits the invasion of *E. faecium* invasion. The presented results suggest that the data-driven approaches are powerful tools to inform the ecology and management of microbial communities.

Microbial communities are constantly exposed to the invasion of exogenous species, which can significantly alter their composition and function[1–4]. The capacity of a microbial community to resist invasion is regarded as an emergent property (i.e., from individual parts to a holistic function) arising from the complex interactions among its constituent species[5]. Theoretical studies have found that communities with higher diversity or stronger interaction strengths among species are more resistant to potential invaders[6–8], attributed to the fact that communities with higher diversity can occupy more niches and provide functional redundancy, making it more difficult for an invading species to establish and thrive.

The role of host-associated microbiota in defending against pathogens has been extensively studied[9–12], particularly in the context of the human gut microbiome. For the human gut microbiome, invasion

---

[1]CAS Key Laboratory of Quantitative Engineering Biology, Shenzhen Institute of Synthetic Biology, Shenzhen Institute of Advanced Technology, Chinese Academy of Sciences, Shenzhen, China. [2]Channing Division of Network Medicine, Department of Medicine, Brigham and Women's Hospital and Harvard Medical School, Boston, MA, USA. [3]Shandong Agricultural University, Tai'an, China. [4]Center for Artificial Intelligence and Modeling, The Carl R. Woese Institute for Genomic Biology, University of Illinois at Urbana-Champaign, Champaign, IL, USA. [5]University of Chinese Academy of Sciences, Beijing, China. [6]These authors contributed equally: Lu Wu, Xu-Wen Wang. ✉e-mail: yyl@channing.harvard.edu; lei.dai@siat.ac.cn

of resident microbial communities can occur when non-resident bacteria from foods and the upper gastrointestinal tract reach the gut ecosystem[13,14]. The resident microbes in the gut ecosystem outcompete and exclude invaders through a combination of mechanisms, such as producing antimicrobial compounds[15,16], competing for nutrients and space[17–20], and modulating the host's immune response[21,22]. However, the composition of the human gut microbiota can vary significantly across individuals[23,24] and over time[25,26]. Dietary shifts, medication, and other environmental factors can greatly alter the composition of the gut microbial community[27,28]. These interpersonal and dynamic variations in the gut microbiome can lead to significantly different colonization outcomes, such as resistance to pathogens[29,30] and probiotics[31,32]. For example, antibiotics treatment often leads to a loss of diversity and promotes the invasion of exogenous species[30,33]. The ability to predict and alter the colonization outcomes (i.e., prevent the engraftment of pathogens and promote the engraftment of probiotics) is critical for personalized microbiota-based interventions in nutrition and medicine.

Despite the accumulating empirical studies, predicting the colonization outcomes in complex communities, such as the human gut microbiome, remains a fundamental challenge due to limited knowledge of interspecies interactions. For a meta-community of $N$ species ($N$ ranges from hundreds to thousands for the human gut microbiome), we would need to isolate and culture all these species (which is already a formidable task) and conduct a considerable number of experiments to map pairwise interactions, not to mention higher-order interactions. Thus, novel approaches are needed to study the ecology of highly complex communities.

The key question is: can we achieve system-level predictions for complex ecological systems without requiring detailed mechanistic information? Colonization outcome can be viewed as a mapping from the community structure of a complex ecological system (i.e., the pre-invasion community profile) to its function (i.e., the post-invasion abundance of the invading species). Recently, the application of data-driven models (machine learning and deep learning) has shown great promise in predicting the emergent properties of complex biomolecules, such as protein structure (mapping from protein sequence to structure)[34], promoter strength (mapping from DNA sequence to function)[35]. While statistical models have been used to decipher microbial interactions in synthetic human gut microbial communities[36] and mouse gut microbiota[30], the use of data-driven models has not received much attention in microbial ecology[37,38].

Here, we proposed a data-driven approach to predict colonization outcomes of exogenous species in complex microbial communities. First, we systematically evaluated the approach using synthetic data generated by classical ecological dynamical models. We found that, with sufficient sample size in training data (on the order of ~$O(N)$), the colonization outcomes (i.e., whether an exogenous species can engraft and what its abundance would be if it can engraft) can be predicted by machine learning models. Then, we generated large-scale datasets with in vitro experimental outcomes of two representative species (*E. faecium* and *A. muciniphila*) colonizing human stool-derived microbial communities. We validated that machine-learning models, including Random Forest and neural ODE, can also predict colonization outcomes in experiments (AUROC > 0.8). Finally, we used the machine learning models to infer species with large colonization impacts and experimentally demonstrated that the introduction of strongly interacting species can significantly alter the colonization outcomes. Our results suggest that the colonization outcome of complex microbial communities can be predicted via data-driven approaches and tunable.

## Results

### The data-driven approach of predicting colonization outcomes for complex microbial communities

Let's consider a meta-community with a pool of $N$ microbial species, denoted as $\Omega = \{1, \cdots, N\}$. Consider a large set of $M$ microbiome samples, denoted as $S = \{1, \ldots, M\}$, collected from this meta-community. A microbiome sample $s \in S$ can be considered as a local community of the meta-community with a subset of co-existing species (Fig. 1A). For a local community $s$, if an exogenous species $i$ (still in the species pool $\Omega$, but not in community $s$) is introduced to community $s$, whether it can successfully colonize the community or not, as well as its post-invasion abundance $x_i$, will depend on the baseline composition of community $s$. For example, it is easier (or harder) for species $i$ to colonize community $s$ if some resident species strongly promote (or inhibit) its growth of species $i$, respectively. Hereafter, we call the community $s$ permissive (or resistant) to species $i$ if species $i$ can (or cannot) successfully colonize community $s$, respectively. If we only have the information about species $i$ and a community $s$, it may seem impossible to accurately predict the colonization outcome without detailed knowledge about microbial interactions. However, if we have access to the data from colonization experiments of many local communities, then, in principle, we can formalize the colonization outcome prediction problem as a machine-learning task that can be solved in a data-driven fashion. To ensure the problem is mathematically well-defined, we must assume that the different local communities in this meta-community share identical assembly rules and microbial interactions[39]. This way, the colonization outcomes of some local communities can be used to train a machine-learning model to predict the colonization outcomes of other local communities.

Consider species-$i$ as the exogenous species to a local community $s$. Note that the baseline abundance of species-$i$ is zero (i.e., $x_i^{(0)} = 0$) in $s$ before the invasion. With some initial abundance, the exogenous species will interact with the resident species in $s$, and its post-invasion steady-state abundance is denoted as $x_i^{(1)}$. We propose to solve the Colonization Outcome Prediction (COP) problem using machine-learning models that treat the baseline (i.e., pre-invasion) taxonomic profile $\mathbf{x}^{(0)}$ as inputs and the steady state abundance of the invasive species $x_i^{(1)}$ as output (Fig. 1B). Mathematically, we intend to learn the mapping from the baseline taxonomic profile of a community $\mathbf{x}^{(0)}$ to the steady state abundance of the invading species $x_i^{(1)}$, i.e., $\varphi : \mathbf{x}^{(0)} \mapsto x_i^{(1)}$. In addition, this mapping could help us infer the impact of each resident species on the colonization of the exogenous species.

We conducted in silico simulations to validate the feasibility of our approach. We generated synthetic data of colonization outcomes using the Generalized Lotka–Volterra (GLV) model with $N = 100$ species in the meta-community (see Methods)[40]. The initial species collection of each sample (i.e., a local community) consists of 30 species randomly drawn from the ($N$−1) species pool (the exogenous species is absent in all the local communities). We generated the baseline profiles of local communities by running the GLV dynamics to a steady state. The exogenous species was then added to each local community, and its post-invasion abundance was obtained by running the GLV dynamics to a new steady state.

We can formalize COP as two sub-problems: (1) *Classification*: predict whether an exogenous species can colonize a local community; (2) *Regression*: predict the steady-state abundance of an exogenous species after colonization. Using the synthetic data generated by the GLV model, we first addressed the *classification* problem, i.e., predicting whether the invading species can colonize a community. We employed three models covering representative categories of machine learning: Logistic Regression, Random Forest classifier, and COP-Neural Ordinary Differential Equations (COP-NODE) classifier (see Methods). We tuned the complexity of the ecological network (i.e., network connectivity) and evaluated the performance of different models at varying levels of the training sample size (Fig. 1C–E). Here, the network connectivity represents the probability of two species in the species pool interacting with each other. As expected, we observed that the predictive performance of machine learning models improved with the number of training samples. For network connectivity $C = 0.3$, we found that the Area Under the Receiver Operating Characteristic

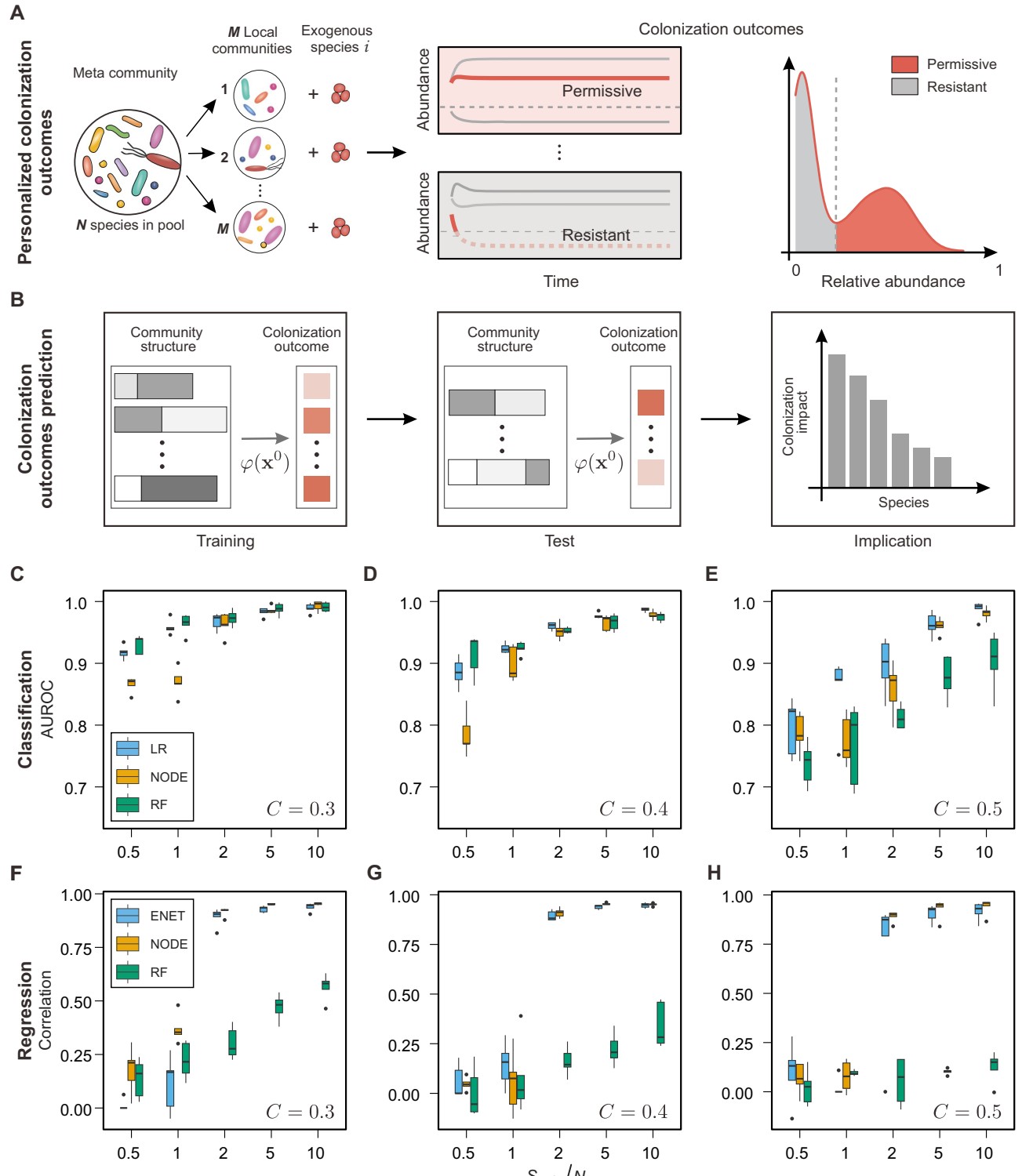

**Fig. 1 | Prediction of colonization outcomes for complex microbial communities via the data-driven approach. A** Each individual's microbiome can be viewed as a local community, a subset of the meta-community of microbial species. For an exogenous species that invades the local communities, its colonization outcome (e.g., permissive or resistant) can be highly personalized, depending on the composition of local communities. **B** Colonization outcome prediction (COP) can be solved by learning the mapping from the baseline taxonomic profile to the post-invasion abundance of the exogenous species (i.e., $\varphi : \mathbf{x}^0 \mapsto \mathbf{x}^1$). **C–E** Evaluation of the data-driven approach in solving the classification task of COP. AUROC of three machine learning models, including Logistic Regression (LR), COP-Neural Ordinary Differential Equations classifier (NODE), and Random Forest classifier (RF). **F–H** Evaluation of the data-driven approach in solving the regression task of COP. Pearson correlation between the true abundance and the abundance predicted by three machine learning models, including Elastic Net Linear Regression (ENET), COP-NODE regressor (NODE), and Random Forest regressor (RF) with network connectivity $C = 0.3, 0.4, 0.5$. Error bars are the 95% confidence interval, the bottom and top of the box are the 25th and 75th percentiles, the line inside the box is the 50th percentile, and outliers are shown as plots.

curve (AUROC, a perfect classifier has AUROC = 1 and AUROC = 0.5 for random guess) of three machine learning models was above 0.9 with training sample size $S_{train} = N$. For higher network connectivity (e.g., $C = 0.4$ and $0.5$), the increased complexity in inter-species interactions rendered the binary prediction of colonization outcomes more difficult. Nevertheless, with a sample size on the order of ~$O(N)$ per colonizing species, machine learning models were able to achieve accurate classification of colonization outcomes in synthetic data (AUROC > 0.8).

Next, we addressed the *regression* problem, i.e., predicting the steady-state abundance of the exogenous species. For the GLV model (with the interaction matrix $A$ being invertible, which is almost surely true for randomly constructed matrices), our analytical derivations discovered a surprisingly simple linear relation between the post-invasion abundance of the exogenous species and the pre-invasion abundance of resident species (Supplementary Text, Fig. S1). Although the linear relation doesn't hold for other dynamical models, it suggests that learning the mapping for COP is feasible by the data-driven approach, and the number of parameters required for fitting the relation is on the order of ~$O(N)$. We employed three machine learning models: Elastic Net Linear Regression (ENET), Random Forest regressor, and COP-NODE regressor (Fig. 1F–H). The predictive performance was evaluated with Pearson's correlation coefficient between the predicted and true abundance (log-transformed), as well as the ratio between the predicted abundance and the true abundance (Fig. S2). We systematically examined the predictive performance of three models at varying levels of network connectivity and training sample size. Similar to the classification problem, we found that increasing network connectivity $C$ rendered the regression problem more difficult. For training sample size $S_{train} = 2N$ or higher, there was a substantial improvement in the quantitative prediction of the post-invasion abundance by ENET and NODE; in contrast, Random Forest had a poor performance at all sample sizes. Finally, we added varying levels of noise in the simulated data to assess the robustness of machine learning models to technical variations (e.g., measurement errors). For both the classification problem and the regression problem, we found that the predictive performance of machine learning models is robust against noise (Fig. S3).

**Generation of human stool-derived in vitro microbial communities with diverse compositional profiles**

To systematically study colonization outcomes in complex microbial communities, we used cultivation of human stool-derived in vitro communities in multi-well plates[41–44] (Fig. 2A, Methods). Briefly, we cultured gut microbial communities derived from 24 donors to reach steady states after five rounds of serial passaging in vitro. To increase the diversity in baseline communities, we treated each donor's sample with a single pulse of 12 antibiotics from different classes (Table S1). After 24 h of antibiotics treatment, in vitro microbial communities were passaged every 24 h with a 1:200 dilution into fresh medium (Fig. 2A). Overall, we obtained more than 300 baseline communities with substantial variation in the compositional profiles at the species level (Fig. 2B, Figs. S4, S5). The compositional profiles of the baseline communities were stable, with around 40 to 120 co-existing species in each community (Fig. S6).

For the invasion experiments, we would introduce an exogenous species into the baseline communities and determine its colonization outcome after 8–10 rounds of serial passaging (Fig. 2C). We conducted a preliminary experiment to investigate the colonization outcome of different exogenous species (Fig. S7). We found that *E. faecium*, *A. muciniphila*, and *Fusobacterium nucleatum* could successfully colonize in some communities at varying levels of post-invasion abundance. In contrast, *Streptococcus salivarius*, *Bifidobacterium breve*, and *Lactobacillus spp.* could not colonize in nearly all the communities we tested. Moreover, vancomycin treatment significantly altered the colonization

outcomes, rendering the gut microbial communities more susceptible to invasion (Fig. S7C). Overall, our results support the use of human stool-derived in vitro communities as a model experimental system for studying colonization outcomes.

**Colonization outcomes of *E. faecium* in human stool-derived in vitro communities**

We selected *E. faecium* as a representative species for colonization experiments in human stool-derived in vitro communities. *E. faecium* is a Gram-positive bacterium that inhabits the gut of humans and other animals. Some *E. faecium* strains have probiotic potential[45], and recent studies suggest that it plays a positive role in cancer immunotherapy[46,47]. On the other hand, some *E. faecium* strains cause opportunistic infections in hospitalized patients with disrupted gut microbiota[48].

We introduced *E. faecium* to ~300 baseline communities (Fig. S8) at a dose of 5% relative to the total abundance of resident species. We passaged all communities for ten rounds to reach the post-invasion steady state (Methods). We observed that the colonization outcomes of *E. faecium* in different communities were persistent during serial passaging (Fig. S8B). In addition, the composition of in vitro communities before and after *E. faecium* invasion is highly reproducible across three replicates (Fig. S9).

We found that *E. faecium* was able to colonize 32% of baseline communities (i.e., permissive), with its post-invasion absolute abundances (estimated by multiplying its relative abundance with the total biomass $OD_{600}$) in permissive communities varying over two orders of magnitude (Fig. 3A). Previous studies suggested that community biomass and diversity are important factors underlying the colonization resistance to exogenous species[49,50]. For example, reduced diversity of the resident community is often linked to pathogen infection in the human gut or other ecosystems[51]. Indeed, we found that the biomass and species richness of the baseline communities exhibited a clear negative correlation with the post-colonization abundances of *E. faecium* (Fig. S10). The diversity of the *E. faecium* permissive communities was significantly lower than the resistant communities (Fig. 3B, C). Furthermore, we observed a significant difference between the composition of *E. faecium* permissive communities and resistant communities (Fig. 3D). The colonization success of *E. faecium* was highly baseline-dependent, with substantial variations across different donors and antibiotics treatments (Fig. S11).

For the regression problem, we need training samples with non-zero post-invasion abundance. Because *E. faecium* only colonized in ~30% of baseline communities in our experiments, the number of samples is insufficient to train the regression models to predict the post-invasion absolute abundance. To predict the binary colonization outcomes (permissive vs. resistant) of *E. faecium*, we employed three machine learning models, including Logistic Regression, COP-NODE classifier, and Random Forest classifier (Fig. 3E–G, Fig. S12AB). For 6-fold cross-validation, we used the communities derived from 20 donors (~240 samples) to train the model and the communities derived from the remaining 4 donors (~60 samples) to evaluate the model. Random Forest classifier displayed the best performance in predicting whether *E. faecium* could successfully colonize based on the species-level community composition (AUROC = 0.86, Accuracy = 0.82), followed by COP-NODE classifier (AUROC = 0.81, Accuracy = 0.81) and Logistic Regression (AUROC = 0.71, Accuracy = 0.75, and the Accuracy of a naive classifier that predicts *E. faecium* cannot colonize in all communities is around 0.7). We also evaluated the performance of machine learning classifiers with a balanced split of training and test samples, showing that Random Forest remains the best classifier (AUROC = 0.81 for *E. faecium* colonization, Fig. S13A). For comparison, we used the community diversity (quantified by the relative species richness) as the only feature to predict colonization outcome (see Fig. S14). Our results indicated that the relative species richness alone

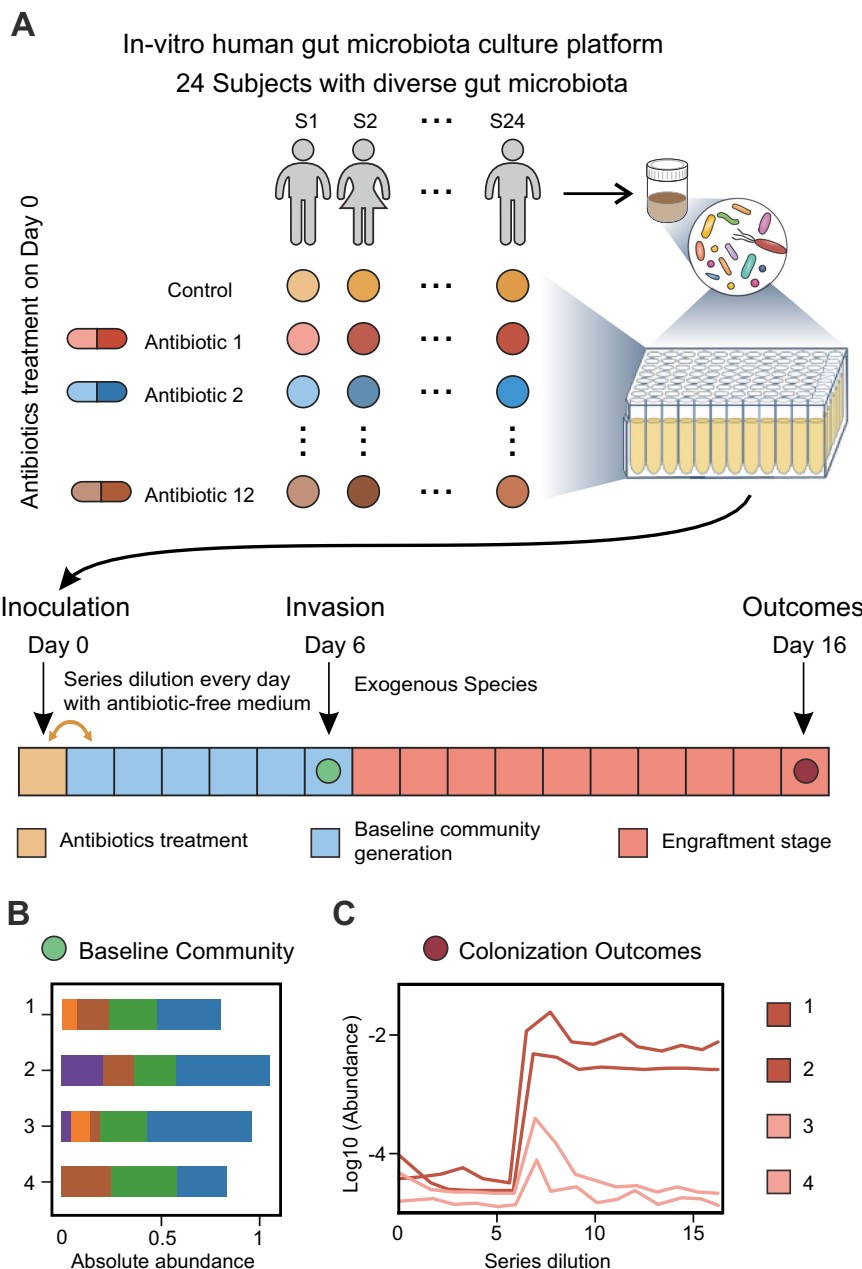

**Fig. 2 | Invasion experiments in human stool-derived in vitro microbial communities. A** Schematic representation of in vitro culture of human stool-derived microbial communities in 96-well plates (Methods). Stool samples from 24 donors were treated with 12 different antibiotics for 24 h. The control group was not treated with antibiotics. All communities were passaged five times to reach a stable state, i.e., baseline community profiles (green dot, schematic cartoon below illustrated an example of in vitro baseline communities with diverse composition). **B** Exogenous species were introduced on Day 6. After 8–10 times of passaging, the end-point community profiles (red dot) were sequenced to determine the colonization outcome (**C**).

can be used as a predictor, but its prediction performance (with average AUROC = 0.78 and Accuracy = 0.32) is worse than elaborate classifiers, e.g., Random Forest using the taxonomic profile (with average AUROC = 0.86 and Accuracy = 0.82). Overall, our colonization experiments of *E. faecium* in complex human gut microbial communities validated that the data-driven approach can solve the classification problem of COP.

### Quantitatively predict the colonization outcomes of *A. muciniphila*

To investigate the generality of our approach, we selected *A. muciniphila* as a second representative species for colonization experiments in human stool-derived in vitro communities. *A. muciniphila* is a Gram-

negative mucin-degrading bacterium that inhabits the human gut. Due to its potential beneficial effects on human health[52–54], *A. muciniphila* is considered a promising probiotic candidate[55]. *A. muciniphila* is found in the gut microbiome of around 30% of adults, and its abundance varies substantially across individuals[56]. Similar to the experimental design of *E. faecium*, we introduced *A. muciniphila* to ~300 baseline communities at a dose of 5% relative to the total abundance of resident species and passaged all communities for eight rounds to reach the post-invasion steady state. The colonization outcome of *A. muciniphila* in different communities was persistent during serial passaging (Fig. S15), and the composition of in vitro communities post *A. muciniphila* invasion is highly reproducible across three replicates (Fig. S16).

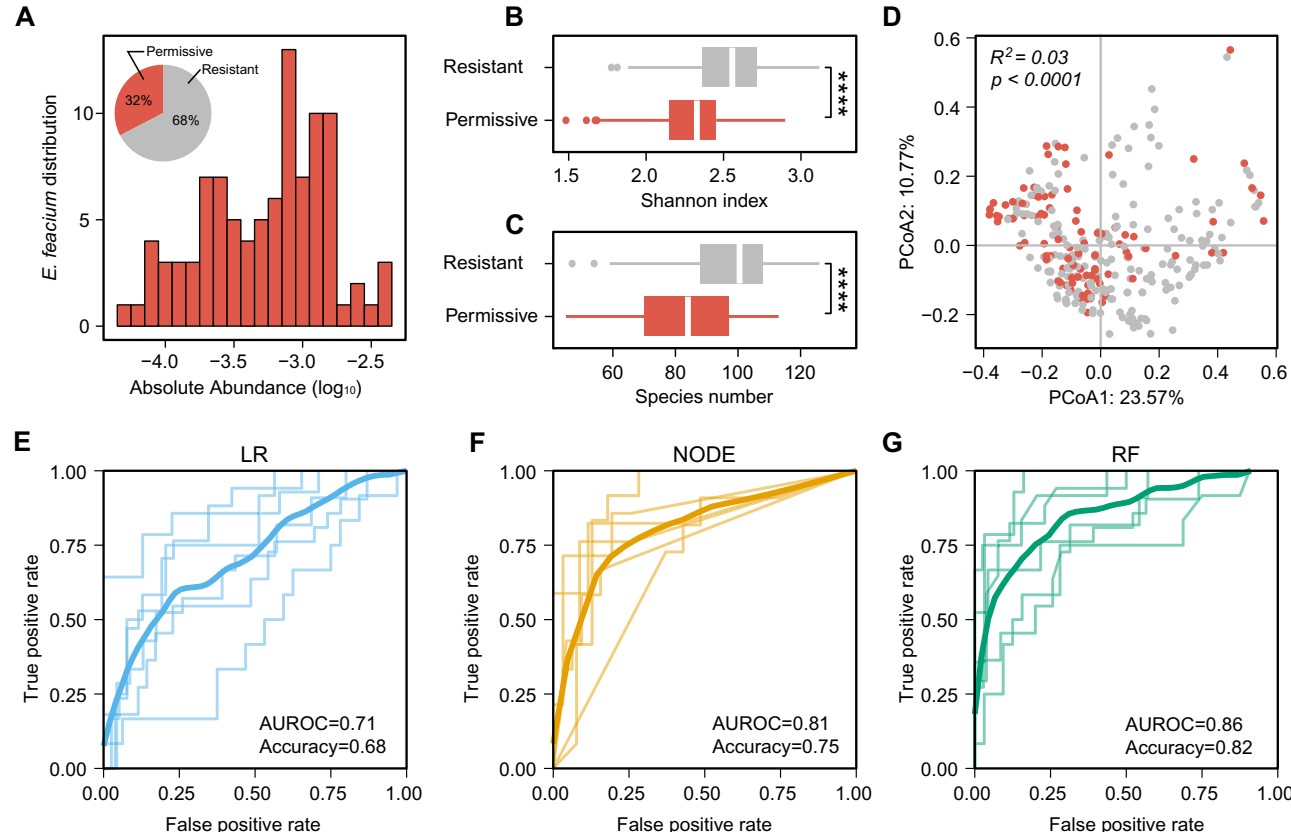

**Fig. 3 | The colonization outcome of *E. faecium* in human stool-derived in vitro microbial communities is baseline-dependent and discriminable. A** The distribution of *E. faecium* colonization outcomes across permissive communities (colored in red). The abundance of *E. faecium* in resistant communities is below the detection threshold and not shown. Inset: Percentage of permissive (red) and resistant communities (gray) based on *E. faecium* colonization outcomes. **B**, **C** Shannon diversity and species richness of *E. faecium* resistant and permissive communities (****$p = 2.902e−10$, Mann–Whitney U-tests, $n = 297$ biologically independent samples). **C** Species richness of *E. faecium* resistant and permissive communities (****$p = 7.655e−13$, Mann–Whitney U-tests, $n = 297$ biologically independent samples). **D** Principal component analysis (PCoA) based on the Bray-Curtis dissimilarity of the compositional profiles of baseline communities. The difference between permissive and resistant communities was significant (PER-MANOVA Adonis test, $R^2 = 0.03, p = 9.999e − 05$). **E–G** ROC curve of machine learning models in binary classification (permissive vs. resistant) of the colonization outcomes of *E. faecium*. For each 6-fold cross-validation (ROC curves shown in a light color), we used the samples from 20 donors to train each model and the samples from the remaining four donors to evaluate the model. The mean ROC curve is shown in dark color. LR: Logistic Regression, NODE: COP-Neural Ordinary Differential Equations classifier, RF: Random Forest classifier. Error bars are the 95% confidence interval, the bottom and top of the box are the 25th and 75th percentiles, the line inside the box is the 50th percentile, and outliers are shown as plots.

Overall, we found substantial variations in the post-invasion steady-state abundance of *A. muciniphila* across different donors and antibiotics treatments (Fig. S17). *A. muciniphila* could colonize in 93.6% of baseline communities (i.e., permissive). For permissive communities, the post-invasion abundance of *A. muciniphila* displayed a bimodal distribution (Fig. 4A). We classified the permissive communities into two subgroups (high vs. low), depending on the post-invasion abundance of *A. muciniphila* (abundance threshold at $10^{-2}$). The Shannon diversity (Fig. 4B) and species richness (Fig. 4C) of the *A. muciniphila* high permissive communities were significantly lower than those of the low permissive communities, and there was a significant difference between the community composition of the two groups (Fig. 4D).

We evaluated the performance of machine learning models in predicting the colonization outcomes of *A. muciniphila*, both qualitatively (classification) and quantitatively (regression). Random Forest classifier displayed the best performance in binary classification (high permissive vs. low permissive of *A. muciniphila*) based on the species-level community composition (AUROC = 0.84), followed by COP-NODE classifier (AUROC = 0.79) and Logistic Regression (AUROC = 0.75) (Fig. 4E–G). To quantitatively predict the post-invasion abundance of *A. muciniphila*, we employed three machine learning models:

ENET, COP-NODE regressor, and Random Forest regressor (Fig. 4H–J, Fig. S18). In comparison to the other two methods, the Random Forest regressor achieved the highest accuracy in quantitative prediction (Pearson's correlation coefficient between the predicted and true abundances $\rho = 0.74, p < 2.2 \times 10^{-16}$) and successfully recapitulated the bimodal distribution in the abundance of *A. muciniphila*. Taken together, we demonstrated the generality of the data-driven approach in predicting baseline-dependent colonization outcomes for complex microbial communities.

## Colonization impact in simulated and experimental communities

Learning the mapping from the baseline taxonomic profile to colonization outcomes can help us infer the impact of each resident species on the colonization of the exogenous species (Fig. 1B). To compute the colonization impact of regression (classification), we can perform a thought experiment by introducing a perturbation in the abundance of the resident species and use the trained machine learning model to predict the new colonization outcome of invading species after the perturbation (Fig. 5A). Negative colonization impact means that a resident species inhibits the colonization of the exogenous species in a given local community. In GLV simulations, while the colonization

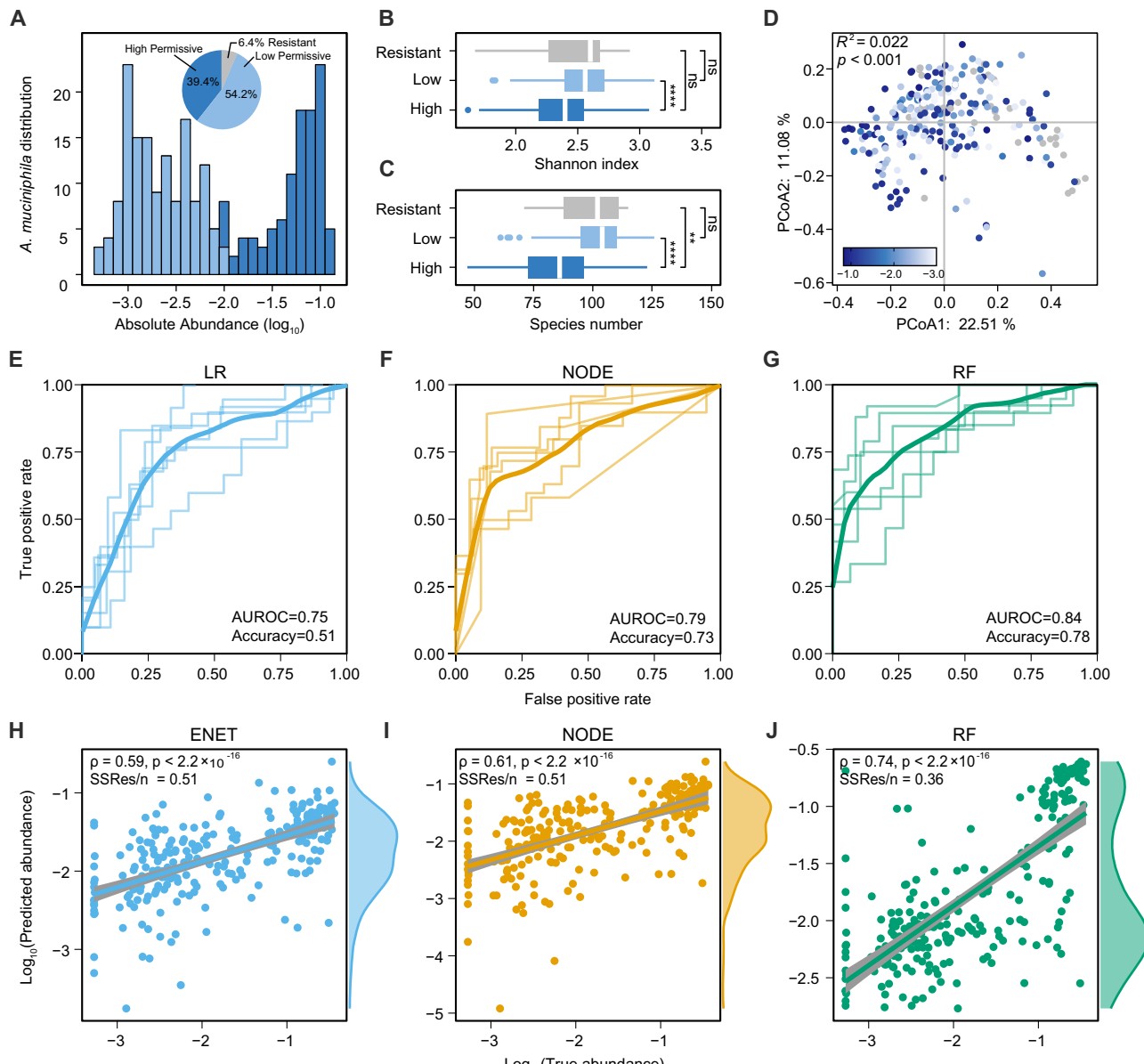

**Fig. 4 | The colonization outcomes of *A. muciniphila* in human stool-derived in vitro microbial communities are quantitatively discriminable. A** The distribution of *A. muciniphila* colonization outcomes across permissive communities. The abundance of *A. muciniphila* in resistant communities is below the detection threshold and not shown. Inset: Percentage of high permissive (dark blue color), low permissive (light blue), and resistant communities (gray) based on *A. muciniphila* colonization outcomes. **B** Shannon diversity of *A. muciniphila* resistant, permissive (low), and permissive (high) communities (ns, not significant, ****$p = 1.644e-15$, Mann–Whitney U-tests, $n = 257$ biologically independent samples). **C** Species richness of *A. muciniphila* resistant, permissive (low), and permissive (high) communities (ns, not significant, **$p = 0.002307$, ****$p = 2.623e-07$, Mann–Whitney U-tests, $n = 257$ biologically independent samples). **D** Principal component analysis (PCoA) plots based on the Bray-Curtis dissimilarity of the compositional profiles of baseline communities. Color of the point showing the abundance of *A. muciniphila* in communities. The difference between highly permissive and lowly permissive communities was significant (PERMANOVA Adonis test, $R^2 = 0.022, p = 9.999e - 05$). **E–G** ROC curve of machine learning models in binary classification (high permissive vs. low permissive) of the colonization outcomes of *A. muciniphila*. For each 6-fold cross-validation (ROC curves shown in a light color), we used the samples from 20 subjects to train each model and the samples from the remaining four subjects to evaluate the model. The mean ROC curve is shown in dark color. ENET: Elastic Net Linear Regression, NODE: COP-Neural Ordinary Differential Equations regressor, RF : Random Forest regressor. **H–J** Pearson's correlation coefficient and the average squared differences between the predicted and the observed abundance (log-transformed values) of *A. muciniphila*. Error bars are the 95% confidence interval, the bottom and top of the box are the 25th and 75th percentiles, the line inside the box is the 50th percentile, and outliers are shown as plots.

impact of resident species was randomly distributed (Fig. 5B), we found that the median colonization impact of a resident species across different local communities was positively correlated to its interaction strength on the exogenous species (Spearman correlation coefficient $\rho = 0.73, p < 2.2 \times 10^{-16}$, Fig. 5C), suggesting that we may use colonization impact to identify strongly interacting species.

We used the Random Forest model to evaluate the colonization impact of all species in human stool-derived in vitro communities on *E. faecium* and *A. muciniphila* (Fig. 5D–G). We inferred that most resident species had a weak negative colonization impact (Fig. 5D, F). Based on the median colonization impact of a certain resident species across different local communities, we identified the top-ranking species with

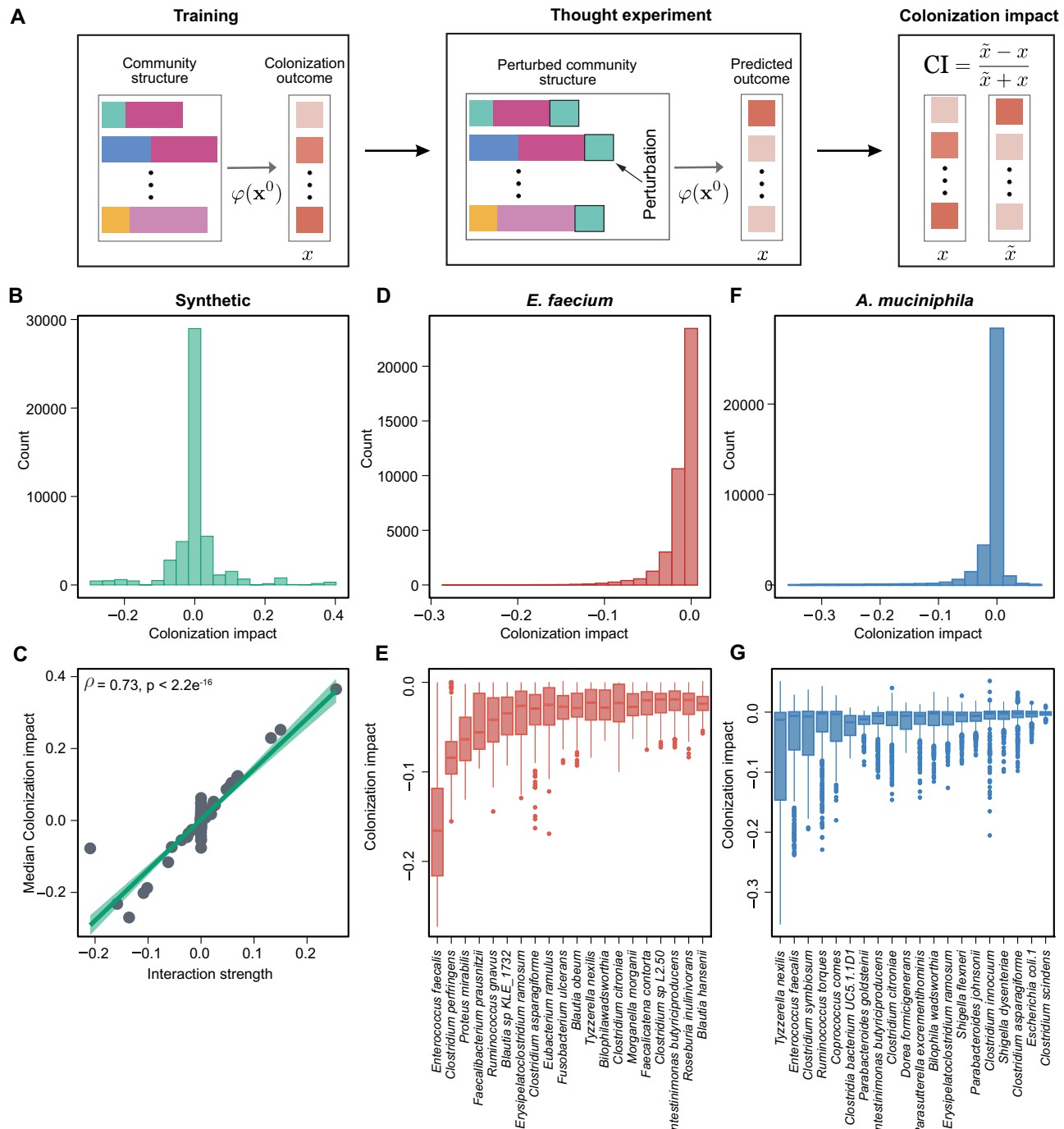

**Fig. 5 | Colonization impact in simulated and experimental communities.**
**A** To compute the colonization impact, i.e., the impact of a resident species on the colonization outcome of the invading species, we first trained the prediction models using all the samples. Then, we performed a thought experiment by introducing a perturbation in the abundance of the resident species and used the trained machine learning model to predict the new steady state abundance of invading species after the perturbation. A negative colonization impact indicates that a resident species inhibits the colonization of the invading species. **B, C** In simulated data, the colonization impact is randomly distributed. The median colonization impact of a resident species across different local communities is positively correlated to its interaction strength on the exogenous species (Spearman correlation $\rho = 0.73$, $p < 2.2 \times 10^{-16}$). Network connectivity $C = 0.3$, $S_{\text{train}}/N = 5$. COP-NODE regressor is used. **D, E** The distribution of colonization impact on *E. faecium*, and the top-ranking species with negative colonization impact (median across different communities, RF classifier). **F, G** The distribution of colonization impact on *A. muciniphila* and the top-ranking species with negative colonization impact (median across different communities, RF regressor). Error bars are the 95% confidence interval, the bottom and top of the box are the 25th and 75th percentiles, the line inside the box is the 50th percentile, and outliers are shown as plots.

negative colonization impact (Fig. 5E, G). Colonization impact on *A. muciniphila* was overall less negative than *E. faecium*, consistent with our observation that human gut microbial communities were more permissive to *A. muciniphila* colonization.

## The impact of strongly interacting species on colonization outcomes
To understand the role of strongly interacting species on colonization outcomes, we systematically studied the impact of *E. faecalis* on the

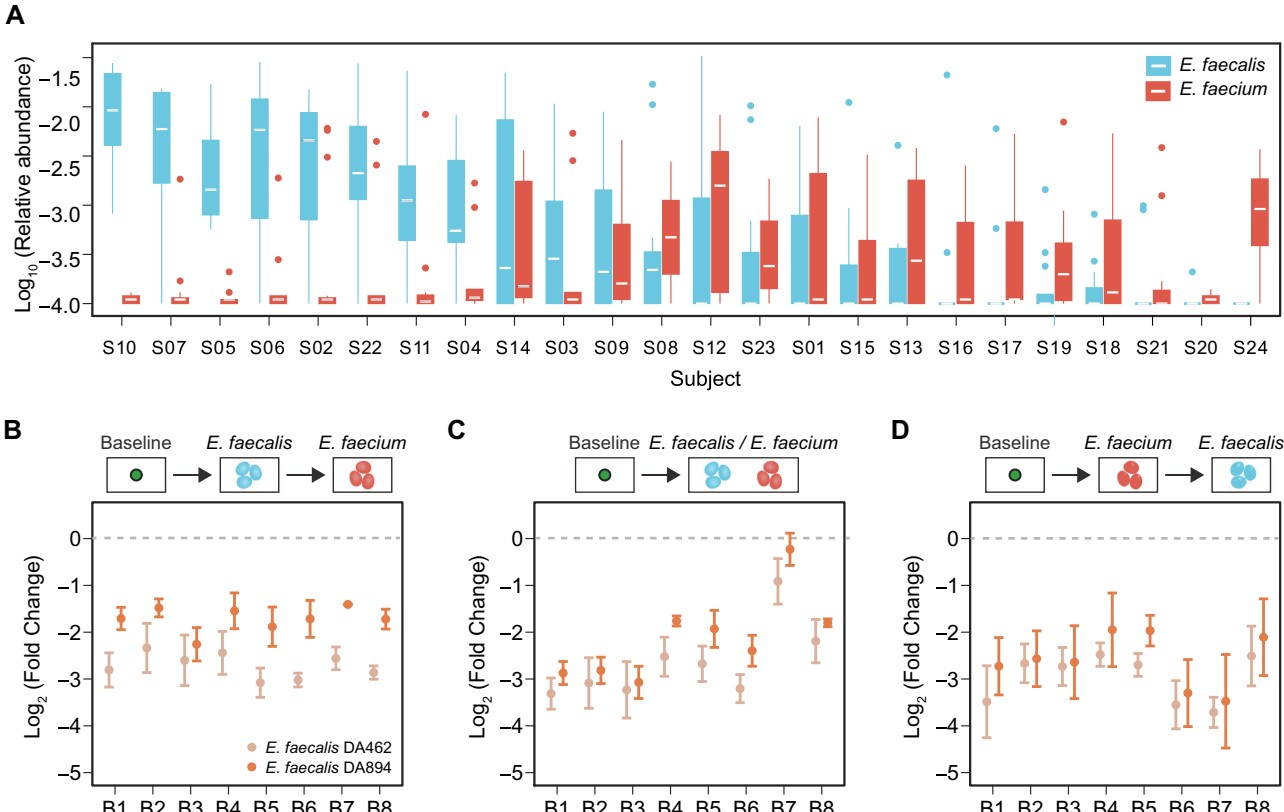

**Fig. 6 | The presence of *E. faecalis* in baseline communities inhibits the invasion of *E. faecium*. A** The post-invasion relative abundance of *E. faecium* (aqua) is negatively associated with the relative abundance of *E. faecalis* (red) across baseline communities derived from different human subjects (labeled as S01 to S24). **B–D** The colonization of *E. faecium* is significantly inhibited by *E. faecalis* across different baseline communities (labeled as B1 to B8). There were three different intervention groups: (1) add *E. faecalis* (or *C. symbiosum*) into the baseline community, followed by *E. faecium* on the next day (**B**); (2) add *E. faecalis* and *E. faecium* on the same day (**C**); (3) add *E. faecium* into the baseline community, followed by *E. faecalis* on the next day (**D**). In the control group, we only added *E. faecium*. After five passages, the end-point abundance of *E. faecium* was measured by qPCR. The fold change in the end-point abundance of *E. faecium* (the intervention group divided by the control group) is lower than 1 (dashed line), indicating that *E. faecalis* inhibits the colonization of *E. faecium*. Two different *E. faecalis* strains, DA462 and DA894, were used. $n = 3$ replicates, the error bars are standard error of means.

colonization of *E. faecium*. *E. faecalis* was inferred to have the strongest colonization impact on *E. faecium* across different baseline communities (Fig. 5E). Besides, we found that *E. faecalis*, as well as other 5 species among the top 20 list (including *Faecalibacterium prausnitzii, Ruminococcus gnavus, Blautia sp., Clostridium sp. L.2.50*, and *Roseburia inulinivorans*) were predicted by both Random Forest (Fig. 5E) and NODE (Fig. S19AB) to have a strong impact on *E. faecium* colonization. In contrast, Logistic Regression classifier did not identify *E. faecalis* as a strong inhibitor (Fig. S19CD), and its predictive performance is substantially worse than Random Forest and NODE (Fig. 3). The colonization impact of those top-ranking species is also consistent with the result of LIME which is a novel explanation technique that explains the predictions of any classifier[57], e.g., the presence of *E. faecalis, F. prausnitzii,* and *Clostridium perfringens* in resistant communities increases the probability that *E. feacium* cannot colonize (see Fig. S19E).

We observed a statistically significant negative correlation between the abundance of *E. faecalis* in baseline communities and the post-invasion abundance of *E. faecium* (Kendall's $\tau = -0.37$, $p = 5.29 \times 10^{-16}$). In particular, baseline communities derived from some donors (e.g., S10, S07) had a high abundance of *E. faecalis* and were resistant to *E. faecium* colonization (Fig. 6A).

We found that *E. faecalis* inhibited the growth of *E. faecium* in pairwise co-culture, either in liquid culture or on agar plates (Fig. S20). Then, we introduced *E. faecalis* into eight human stool-derived in vitro communities that were permissive to *E. faecium*

invasion, using three different types of interventions (Fig. 6B–D, Fig. S21A): (1) add *E. faecalis* into the baseline community, followed by *E. faecium* on the next day; (2) add *E. faecalis* and *E. faecium* on the same day; (3) add *E. faecium* into the baseline community, followed by *E. faecalis* on the next day. In the control group, we only added *E. faecium*. In all three intervention groups, the colonization of *E. faecium* was significantly inhibited by *E. faecalis* across different baseline communities. Also, the inhibitory effect was consistent for two different *E. faecalis* strains isolated from human stool samples (Methods). In comparison, *Clostridium symbiosum*, a species predicted to have a neutral impact, did not alter the colonization of *E. faecium* (Fig. S21B).

Finally, we explored if the strong inhibition of *E. faecalis* on *E. faecium* could be shaping their distribution in the human gut via priority effects, i.e., the gut microbiome colonized with *E. faecalis* becomes resistant to *E. faecium*. We performed metagenomic sequencing of ~120 healthy volunteers in the SIAT cohort (Methods), whose samples were used to derive the in vitro communities and isolate the *Enterococcus* strains in this study. Indeed, there was a statistically significant negative correlation between the relative abundance of *E. faecalis* and *E. faecium* in the SIAT cohort (Kendall correlation $\tau = -0.36, p = 0.0044$, Fig. S22A). A similar pattern was observed in gut metagenomic samples of four independent cohorts (Kendall correlation $\tau = -0.36, p = 5.439 \times 10^{-15}$, Fig. S22B).

Overall, our experimental validations and analysis suggest that data-driven models can infer species with strong colonization impact

and guide the modulation of resident communities to alter the colonization outcomes of exogenous species.

## Discussion

Here we proposed and systematically validated a data-driven approach to predict colonization outcomes of exogenous species, providing a powerful tool to inform the management of complex ecosystems. Pairwise co-culture[50,58–60] and synthetic communities[17,36,61,62] have been widely used to study the ecology and function of microbial communities. These experiments require the isolation and cultivation of individual species, thus are often limited to simple communities. In comparison, our approach is based on sampling an ensemble of complex communities (~100 species, Fig. S4) and using the sampled communities to infer the mapping between community composition and colonization outcomes[63] by assuming the compositional profiles represent steady states of the local communities. We demonstrate that the data-driven approach enables accurate function prediction and system-level understanding of complex microbial communities.

Understanding the colonization resistance of complex communities is a fundamental question in ecology. In our invasion experiments (~300 local communities and two different exogenous species), we found that resistance to exogenous species was positively correlated to community diversity, supporting the view that colonization resistance is an emergent property of complex communities[8]. Community diversity (i.e. species richness) alone can provide a reasonable degree of predictive accuracy (Fig. S14). However, it is less effective compared to utilizing full taxonomic profiles. While most resident species had a weak negative impact on the colonization of exogenous species, we identified *E. faecalis* as a strong inhibitor of *E. faecium*. We validated that introducing strongly interacting species into baseline communities can alter the colonization outcomes. It should be noted that the colonization impact is dependent on the community context (Fig. 5), because it takes into account both direct and indirect effects on the invading species[64], as well as potential higher-order interactions[65,66]. Previous studies have shown that strongly interacting species can lead to priority effects[67], with important implications for community assembly in the infant gut microbiome and the formation of community types[68,69]. Moreover, strongly interacting species can be used to modulate the resident communities to prevent the colonization of pathogens[16] or facilitate the colonization of beneficial microbes (e.g., probiotics, crop fertilizers)[70].

Our results suggest that the colonization resistance of microbial communities is predictable and tunable via the data-driven approach, given that training data size is sufficient (on the order of ~$O(N)$)[71]. For synthetic data generated by the classical GLV model in community ecology, we did see that the simple models, e.g., linear regression and ENET work well for both classification problems (Fig. 1C–E) and regression problems (Fig. 1F–H). However, those simple statistical models did not work well in predicting the colonization of *A. muciniphila* (Fig. 4H). We anticipate that real microbial communities are more complicated than the simple GLV model (which only includes pair-wise inter-species interactions). Sophisticated machine learning models may have to be leveraged to predict colonization outcomes for complex communities. We anticipate that more training samples are required if high-order interactions are considered. However, those high-order interactions might be weak and do not significantly impact the prediction, as the community-function landscapes display a low degree of ruggedness[72]. The high-throughput cultivation of gut microbial communities in vitro provides a powerful approach to studying the human gut microbiome[44,73]. In our experiments, the number of species in the meta-community was ~160, and we profiled ~300 baseline communities for proof-of-concept validation. Meeting the sample size requirement for gnotobiotic plants is feasible[74,75]. However, it could be challenging to gather sufficient training data for gnotobiotic animals and human cohort studies, depending on the

complexity of the meta-community. In addition to data size, another critical concern is the technical variability in large-scale experiments[76]. In future studies, experimental workflows can be automated to minimize technical variability and ensure data quality for training machine learning models.

Our study has several limitations. First, we did not account for potential variations at the strain level[77]. Previous studies have shown that the strength of interspecies interactions can vary across different strains, such as the inhibition of *Klebsiella pneumoniae* by *Klebsiella oxytoca*[16,78]. We also observed strain-level variations in the inhibition of *E. faecium* by *E. faecalis* (Fig. 6), and the underlying mechanism remains to be elucidated. Second, our invasion experiments in vitro did not reflect host-mediated interactions, which also contribute to colonization resistance in vivo[22]. Nevertheless, the higher permissiveness to *A. muciniphila* than *E. faecium* in human gut microbial communities in vitro is consistent with the higher prevalence of *A. muciniphila* in metagenomic samples[56,79]. Third, we assumed that there was a single post-invasion steady state in simulated and experimental communities. The colonization outcomes may be influenced by multi-stability in microbial communities, e.g., successful colonization depends on the initial abundance of the invading species[80,81]. Fourth, in our in vitro experiments, we found that *A. muciniphila* was able to stably colonize in the majority of stool-derived communities with relatively high abundance. It is known that mucin is the preferred nutrient source of *A. muciniphila*[82], so it would be interesting to study to which degree the colonization of *A. muciniphila* depends on the mucin concentration provided in the medium. Lastly, while the data-driven framework can be generalized to different scenarios, the machine learning models must be re-trained when the environmental condition changes. In contrast, mechanism-based models can better deal with changes in conditions (e.g., pH, nutrient level).

We noted a potential difference between the GLV simulated data and experimental data: the exogenous species may already be present in the stool-derived communities, but its steady-state abundance was below the detection threshold. In this scenario, the introduction of the exogenous species (~5% of community biomass) may provide a growth boost (e.g., via some density-dependent mechanism) and enable the species to co-exist with other species at a higher steady-state abundance (i.e., multiple stable states). Moreover, there is a discrepancy between the performance of Random Forest in gGLV simulated data and real data. Potential explanations include: (1) the dynamics of the GLV model may be different from that of experimental communities. For instance, when the GLV model has globally stable equilibria, the final state is solely determined by the species collection. (2) the distribution of interspecies interaction strength used in the GLV model may differ from experimental communities. In experimental communities, a few strongly interacting species may dominate the contribution to the colonization resistance of exogenous species. In contrast, in simulated data, the contribution is more evenly distributed among resident species.

Our data-driven approach is independent of any dynamics model to predict colonization outcomes of exogenous species for complex microbial communities without detailed knowledge of the underlying ecological and biochemical process. We anticipate that the data-driven approach can be generalized to predict and engineer the function of microbial communities (i.e., mapping from community composition to function)[37,71,83,84]. Similarly, this approach can be used to predict the response of microbial communities to various types of perturbations (i.e., mapping from community composition to the shift in composition/function), such as the baseline-dependent response of the human gut microbiome to prebiotics, food additives, etc., refs. [85,86]. In parallel to the breakthroughs in predicting the properties of complex biomolecules, we envision that the data-driven approach will lead to a paradigm shift in studying the stability and function of complex ecological systems and guide important applications in healthcare (e.g.,

personalized nutrition based on the human gut microbiome) and agriculture.

## Methods

### Collection and preservation of human stool samples

All human participants at SIAT (referred to as "SIAT cohort") signed the informed consent form in the present study which was approved by the Shenzhen Institute of Advanced Technology, Chinese Academy of Sciences (SIAT-IRB-200315-H0438). Stool samples were collected from healthy human donors and were immediately transferred to an anaerobic workstation (85% $N_2$, 10% $H_2$ and 5% $CO_2$, COY). 10 g of each stool sample was suspended into 50 mL 20% glycerol (v/v, in sterile phosphate-buffered saline, with 0.1% L-cysteine hydrochloride), homogenized by vortexing, and then filtered with sterile nylon mesh to remove large particles in fecal matter. Aliquots of the suspension were stored in sterile cryogenic vials and frozen at −80 °C for long-term storage until processing for DNA extraction and culturing so that the stool-derived community could be revived (thawed) for repeatable experiments.

### Cultivation of human stool-derived in vitro communities

20ul stool slurries aliquot stocks were inoculated into 980 μL medium containing antibiotics in triplicate into 96 deep-well plates (PCR-96-SG-C, Axygen) for static culturing at 37 °C for 24 h in the anaerobic workstation. The concentration for each antibiotic was evaluated as described in the SI method. The medium (MiPro) used for in vitro culture was modified from previous studies, which comprises: peptone water (2.0 g/L, CM0009, Thermo Fisher), yeast extract (2.0 g/L, LP0021B, Thermo Fisher), L-cysteine hydrochloride (1 g/L), Tween 80 (2 mL/L), hemin (5 mg/L), vitamin K1(10 μL/L), NaCl (1.0 g/L), $K_2HPO_4$ (0.4 g/L), $KH_2PO_4$ (0.4 g/L), $MgSO_4 \cdot 7H_2O$ (0.1 g/L), $CaCl_2 \cdot 2H_2O$ (0.1 g/L), $NaHCO_3$ (4 g/L), porcine gastric mucin (4 g/L, M2378, Sigma-Aldrich), sodium cholate (0.25 g/L) and sodium chenodeoxycholate (0.25 g/L)[87]. After 24 h of antibiotics treatment, in vitro microbial communities were passaged every 24 h with a 1:200 dilution into fresh medium using the automated 96-format Thermo Scientific™ ClipTip™ (Thermofisher) pipette (every 24 h, 5 μL of this saturated culture was transferred into 995 μL of fresh medium). After 5 days of passaging, 500 μL of the cultures were mixed with 500 μL sterile 40% glycerol (v/v, in sterile phosphate-buffered saline, with 0.1% L-cysteine hydrochloride) in crimp vials, sealed, and stored as baseline communities at −80 °C for further usage and long-term storage. After each transfer, the remaining samples were centrifuged to remove the supernatant, and the pellets were stored at −80 °C with a plastic seal until DNA extraction. The in vitro microbial community biomass was evaluated by measurement of optical density ($OD_{600}$) with an Epoch 2 plate reader (BioTek) after 24 h of incubation.

### Generation of baseline communities with diverse taxonomic profiles

To examine if in vitro stool-derived communities can reach stable states and display diverse compositions, we collected stool samples from healthy donors and grew them in MiPro medium, which has shown its capability in capturing and maintaining the diversity of in vitro stool-derived communities[42,87,88]. We inoculated the stool aliquots into 96-well plates with growth media and incubated them in an anaerobic workstation in triplicate, passing them every 24 h with a 1:200 dilution. The microbial communities were assessed by shallow metagenomic sequencing, which is a cost-effective method for characterizing species-level composition of microbiota samples[89]. We collected time-series data to examine the dynamics of community establishment on the in vitro platform. The metagenomic analysis revealed that, after an initial period of approximately four days, the composition profiles of almost all in vitro communities reached a stable and reproducible steady state. Our analysis also showed that the

stool-derived in vitro communities were highly complex in their compositions and could retain personalized gut microbiota variation, as evidenced by species-level time-series compositions of 4 representative communities derived from 4 donors over ten rounds of in vitro passaging in MiPro (Fig. S6A, B).

From the fecal samples of SIAT cohort, we selected 24 donors in which *E. faecium* and *A. muciniphila* were not detected by metagenomic sequencing. To increase the diversity in baseline communities, we treated each donor's sample with 12 antibiotics from different classes[90] (Fig. S4). Those stool-derived communities were treated with antibiotics for 24 h on Day 0 (i.e. a single pulse). Afterwards, the communities were passaged five times (from Day 1 to Day 6) in antibiotic-free medium to reach a stable state before introducing the exogenous species. Different antibiotic classes target distinct spectra of bacteria, leading to a remodeling of the community in different directions[90]. We selected antibiotics from different classes as described in the EUCAST databases[91]. The optimal concentrations of the antibiotics were determined based on a previous study that evaluated the activity spectrum of antibiotic classes on human gut commensals[90]. We tested at least three different concentrations for each antibiotic and evaluated the optimized dose based on its ability to partially inhibit (50–80%) the overall growth of stool-derived bacteria as measured by $OD_{600}$ after 24 h of incubation. To ensure reproducibility, we screened at least three different stool aliquot stocks as biological duplicates for each antibiotic. We measured the $OD_{600}$ of each well every 30 min using an Epoch 2 plate reader (BioTek) and collected growth curves up to 24 h.

### Bacterial strains

*Enterococcus faecium*, *Enterococcus faecalis*, *Clostridium symbiosum*, *Streptococcus salivarius,* and *Bifidobacterium breve* strains were isolated from fecal samples of SIAT cohort. Taxonomy of isolates from SIAT cohort was confirmed by whole genome sequencing. Genome sequences have been deposited in PRJEB60398 (see "Data availability"). *Lactobacillus plantarum HNU082*[92], *Lactobacillus paracasei HNU312*[93] was provided by Prof. Jiachao Zhang from Hainan University. *Akkermansia muciniphila* (ATCC BAA-835) and *Fusobacterium nucleatum* (ATCC 25586) were purchased from ATCC.

### Profiling the colonization outcomes of different exogenous species

We conducted a preliminary experiment to investigate the colonization outcome of gut microbial communities to different exogenous species (Fig. S7), including: *E. faecium, A. muciniphila*[94], *F. nucleatum, S. salivarius, B. breve,* and *Lactobacillus spp.* (*L. plantarum HNU082* and *L. paracasei HNU312*). We identified 12 stool samples from healthy donors in which the selected invader species were undetectable in the microbiota. We then cultured the stool samples in vitro and exposed them to antibiotics before introducing the exogenous species (~5% of total biomass, approximately $10^6$ CFUs for each well) into the community. We used shallow metagenomic sequencing to monitor the time-series and final community composition.

### Invasion experiments of *E. faecium* and *A. muciniphila*

To conduct invasion experiments, frozen stocks of *E. faecium* (strain SIAT_DA797) and *A. muciniphila* (strain ATCC BAA-835) were grown anaerobically in BHI and mGAM at 37 °C, respectively, until stationary phase. In vitro microbial baseline communities, stored at −80 °C, were thawed and revived by adding 20 μL of the stocks to 980 μL of MiPro medium in deep-well plates. After incubation for 24 h at 37 °C, community biomass was measured by $OD_{600}$, and 5 μL of the saturated cultures were diluted into 1 mL of fresh MiPro in a new plate. Each well was invaded with the respective amount of *E. faecium* or *A. muciniphila*, with biomass representing 5% of the inoculated communities' average biomass. The inoculum was passaged every 24 h of incubation,

with a 1:200 dilution into fresh medium for 8–10 passages until the community reached a steady state (10 passages for *E. faecium*, 8 passages for *A. muciniphila*, based on data from Fig. S6). After each passage, the remaining samples were centrifuged to remove the supernatant, and the pellets were stored at −80 °C with a plastic seal in plate until DNA extraction.

### Metagenomic sequencing and taxonomic profiling

DNA was extracted from 200 mg of stool samples using the QIAamp Power Fecal Pro DNA Kit (Qiagen) according to the manufacturer's instructions. For stool-derived in vitro-cultured samples, 500 uL of cultured samples were used for DNA extraction with the DNeasy UltraClean 96 Microbial Kit (Qiagen) using an automated protocol at Tecan Freedom EVO 200. The Hieff NGS® OnePot II DNA Library Prep Kit for Illumina® (Yeasen) was used for library preparation, following the manufacturer's instructions. The resulting library DNA was cleaned up and size-selected with Hieff NGS® DNA Selection Beads (Yeasen), and quantified using the dsDNA High Sensitivity kit on a Qubit (Thermo Fisher). Libraries were further pooled together at equal molar ratios, and the purity and library length distribution were assessed using Bioanalyzer High Sensitivity DNA Kit (Agilent). Sequencing was performed on the Illumina HiSeq X Ten system (150 bp paired-end reads; Annoroad Gene Technology Co.), with a target sequencing depth of 0.3 Gbp raw data per sample, as recommended by previous studies[89].

Samples with fewer than $10^5$ clean reads were excluded from downstream analysis. Prior to analysis, reads were trimmed using the following criteria: (1) Removing reads with more than 50% of the base below quality score 19; (2) Removing reads with more than 5% of the base being N; (3) Discarding paired-end reads if either of the paired reads did not meet the above criteria. Microbial community composition from metagenomic sequencing data was generated using the SHOGUN pipeline and the RefSeq database version 82, as described in previous studies[89,95]. Species-level abundance profiles were filtered by using a relative abundance threshold of 0.0001 (0.001) for all taxa in colonization prediction of *E. faecium* (*A. muciniphila*), and those low-prevalence taxa (present in less than 20% samples) were further filtered to reduce the feature number. The colonization outcomes were evaluated based on the invader's absolute abundance in the community, which was estimated by multiplying the relative abundance and the $OD_{600}$ value ($OD_{600}$ × relative abundance). To ensure repeatability, samples with Pearson correlation below 0.8 among replicates were excluded from COP analysis. This resulted in the exclusion of 1.8% of samples for *E. faecium* and 1.3% for *A. muciniphila*.

### Quantification of the relative abundance of *E. faecium* and *A. muciniphila* by metagenomic sequencing

To confirm the accuracy of shallow metagenomic sequencing in quantifying the relative abundance of *E. faecium* and *A. muciniphila*, a spike-in experiment was conducted (Fig. S23A). In this experiment, a predefined amount of bacterial DNA from the target species was added to a metaDNA sample extracted from an in vitro community derived from human stool. This metaDNA sample was used as the background, since it has been previously sequenced and did not contain the target species. The spike-in DNA of the target species (*E. faecium* or *A. muciniphila*) was 1:10 diluted for eight times and was added to the microbial metaDNA to a mixed DNA sample (5 μL of target species DNA into 30 ng of microbial metaDNA). Three replicates were made for each sample. The mixed DNA was then used for library construction and metagenomic sequencing. By comparing the detected relative abundance generated by shallow metagenomic sequencing with the expected abundance, the accuracy and sensitivity of our workflow were determined. The detection threshold of *E. faecium* is 0.0001 (Fig. S23B) and the detection threshold of *A. muciniphila* is 0.001 (Fig. S23C). Our results showed that the quantification of the relative

abundance of the two target species using the shallow metagenomic sequencing pipeline is accurate and reproducible.

### Statistical analysis

Statistical details for each experiment are indicated in the figure legends. Pearson correlation coefficients and the p-values for testing replicates communities' composition correlation were calculated on $\log_{10}$(relative abundance). Kendall correlation coefficients and the p-values for testing *E. faecium* and *E. faecalis* abundance correlation were calculated on $\log_{10}$(relative abundance). Alpha diversity of the community was calculated on species profile using the observed species richness and Shannon index. The composition of microbiota and variations in colonization outcomes between communities were analyzed by performing PCoA using the Bray-Curtis dissimilarity metric on the species-level abundance profile. Similarities among groups were determined by permutational multivariate analysis of variance (PERMANOVA, Adonis test) based on the Bray-Curtis dissimilarity[96], with 999 permutations used to test the significance. These analyses were conducted using the vegan[97] package (version 2.6–4). Non-parametric Mann–Whitney U-test were used to conduct pairwise comparisons between two groups[98]. *P* values of less than 0.05 were considered as statistically significant, as indicated in the figures (ns, not significant, $*p < 0.05$, $**p < 0.01$, $***p < 0.001$, $****p < 0.0001$). Data analysis and plotting was performed in R version 4.1.2 and R studio version 2022.12.0 + 353 using the packages dplyr, ggpubr, vegen, and ComplexHeatmap.

### Reporting summary

Further information on research design is available in the Nature Portfolio Reporting Summary linked to this article.

## Data availability

All sequencing data generated in this study are available from European Nucleotide Archive (ENA) under study accession number PRJEB60398. Sample accession code, metadata and related source data are provided as a Source data file with this paper. Source data are provided with this paper.

## Code availability

The code for simulations and data analysis is available at https://github.com/spxuw/COP.

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

## Acknowledgements

We thank Na Li and volunteers at Shenzhen Institute of Advanced Technology (SIAT) for stool sample collection. We thank Zepeng Qu and Zhenkun Zhang for isolating human gut bacterial strains. We thank Prof. Jiachao Zhang at Hainan University for providing probiotic strains. We thank Huaijie Hao and Yan Tan at Xbiome for providing help with the anaerobic workstation. We thank Shenzhen Infrastructure for Synthetic Biology and Chaobi Lei for providing support in DNA extraction. We thank Zheng Sun, Chen Liao, Hongbin Liu, Lanxiang Wang, and colleagues at SIAT for valuable discussions. L.D. acknowledges support from the National Key R&D Program of China (2019YFA0906700), the National Natural Science Foundation of China (31971513), and Shenzhen Key Laboratory for the Intelligent Microbial Manufacturing of Medicines

(ZDSYS20210623091810032). Y.-Y.L. is supported by grants R01AI141529, R01HD093761, RF1AG067744, UH3OD023268, U19AI095219 and U01HL089856 from the National Institutes of Health, USA; a pilot grant from the Biology of Trauma Initiative of Broad Institute, USA; and the Office of the Assistant Secretary of Defense for Health Affairs, through the Traumatic Brain Injury and Psychological Health Research Program (Focused Program Award) under award no. (W81XWH-22-S-TBIPH2), endorsed by the Department of Defense, USA. L.W. acknowledges support from the National Natural Science Foundation of China (32100089). X.W.W. acknowledges the funding support from the National Institutes of Health (K25HL166208).

## Author contributions

L.D., Y.Y.L., L.W., and X.W.W. conceived the presented idea, designed the project, interpreted the results, and wrote the manuscript. X.W.W. implemented in silico simulations, machine learning models, and analytical derivations. L.W. and Z.T. performed the in vitro experiments. Y.Z. assisted in experiments. L.W. analyzed the experimental data. T.W. discussed the results and revised the manuscript. W.Z. assisted in data analysis and interpretation. All authors approved the final manuscript. L.D. and Y.Y.L. provided supervision and resources for this study.

## Competing interests

The authors declare no competing interests.
