## [Peer Review File · Nature Communications]

Reviewers' Comments:

Reviewer #1:

Remarks to the Author:

In their work, Wu et al. investigated how easily *E. faecium* and *A. muciniphila* could invade 300 stool communities derived from 24 healthy donors supplemented with 12 different antibiotics. They saw that invasion success depended on stool source, antibiotic treatment, community diversity and biomass. They then predicted invasion success with three supervised classification approaches and found that *E. faecalis* had a large negative impact on the colonisation of *E. faecium*. Finally, they confirmed the relationship between *E. faecium* and *E. faecalis* experimentally. This is an important work, but I see a number of major issues.

Major comments

The stool communities are treated with 12 different antibiotics. Thus, the authors cannot describe communities as permissive or resistant since it is not clear whether the community composition per se or the antibiotic treatment hampered colonisation. In this context, we need to know how sensitive *E. faecium* and *A. muciniphila* are to the 12 antibiotics tested. Perhaps this sensitivity is as good as or even a better predictor of invasion success than the composition of the target communities. If this turns out to be the case, the entire point of this study is undermined and the authors would need to demonstrate that machine learning approaches can predict invasion success on untreated stool samples.

The authors predict *E. faecium*/*E. faecalis* as a strongly interacting pair based on the importance of *E. faecalis* in Random Forest classification. Do the other two classification approaches also point to *E. faecalis*? This could be explored e.g. with LIME (<https://arxiv.org/abs/1602.04938>). There may be other strongly interacting species that were either not detected by the machine learning approaches or that were hidden by the effect of the antibiotic.

L. 227-228: 240 training versus 60 test samples from 20 versus 4 donors is a very unbalanced design with a risk of overfitting. Does classification still work when training and testing are more balanced?

L. 338-340: "We demonstrate that the data-driven approach enables accurate function prediction and system-level understanding of complex microbial communities."

This is a big claim. Which systems-level understanding did the authors gain? They did not learn the mechanism behind the interaction between *E. faecalis* and *E. faecium*. The advantage of knowing mechanisms is that the model is better able to deal with changes in conditions, such as lower pH or different nutrient levels. A machine learning approach may work in a given situation but will have to be retrained when circumstances, such as the medium, change. This weakness of the data-driven approach is not discussed.

The gLV simulations deviate from the real case in an important point: the exogenous species may have been already present in the stool communities, perhaps below detection level. In that case, not the colonisation itself is of interest but the boost given to the "exogenous" species (which may be due to some density-dependent mechanism). This issue needs to be discussed. Another problem is that no noise was added to the simulated data. However, sequencing comes with large technical variation, so it is important to assess the robustness of the machine learning tools to such variation.

L. 165-166 and elsewhere: when assessing prediction accuracy for the abundance of a species, the Pearson correlation is not ideal, since it does not consider abundances themselves, only trends. It is more informative to report the average ratio between predicted and observed abundance here. When comparing predicted and observed community compositions (e.g. Fig S7), Pearson is also not appropriate and better replaced by Bray Curtis.

Is the OD correlated to actual measurements of biomass (gram dry weight) or to cell densities derived from flow cytometry?

Minor comments

How variable are stool communities coming from the same donor and with the same treatment? Is that variability exceeded by the variability across treatments (per donor)?

Why do detection thresholds for *E. faecium* and *A. muciniphila* differ by an order of magnitude?

Another limitation to be discussed is the presence of mucin in liquid form. I assume this was necessary to enable the growth of *A. muciniphila*, but it deviates from the biology of the intestinal system, where *A. muciniphila* prefers to colonise (solid) mucin and other gut bacteria prefer the lumen. It would be of interest to find out to what extent *A. muciniphila*'s ability to colonise the tested communities depends on the mucin concentration in the medium.

Reviewer #2:

Remarks to the Author:

This is an exciting paper using machine-learning approaches to predict the ability of a species to invade an existing community, as well as the final abundance of the invading species. Then they evaluate the same methods with simulated data (using a generalized Lotka-Volterra model) and with real communities using replicated self-assembly communities and antibiotics to create variation in community contexts.

This paper is an important contribution to the field. My main concern is that in the description of the results, some negative or less interesting results are not described. I think the paper is really good but omitting or glossing over negative results can generate distrust in the reader. In particular:

I have a few large comments that are more discussion points than criticisms of the paper:

1. To what extent maybe simpler statistical models (including LASSO regression) still predict the data? Are there less data-intensive methods that could still provide accurate predictions?
2. Why does random-forest performs poorly in the regression task with the Lotka-Volterra simulations but it is the best-performing algorithm in real communities? Any hypotheses?
3. Emphasize in the discussion that these predictions require communities in a steady state (or to what extent are the methods robust to this assumption).

Smaller comments:

L100- I wonder if there is a way to map these variables in figure 1 for extra clarity.

L170 - State in the Lotka-Volterra results that random forest performs poorly at all sizes.

L230. Why are regression results not presented for *E. faecium*? in the main text and figure 3?

Figure 1. Make invading species clearer, maybe the lines of other species in the community should be shades of gray.

Figure 2B. Is this real data? Which communities are these?

Figure 6 - Having the legend on each side of the dashed line is confusing, makes it seem like one strain dominates on one side of the line vs. the other (until carefully reading the caption).

Reviewer #3:

Remarks to the Author:

In this manuscript, Wu & Wang et al study the problem of predicting whether a given bacterial species will successfully invade a larger microbial community. They utilize a "data driven" approach, in which the observed colonization outcomes of a given invader across a large number of "training" communities are used as input data to fit machine learning models to extrapolate colonization outcomes to other "test" communities. They demonstrate this approach on simulated data from Generalized Lotka-Volterra models, and show that several machine learning models have reasonable accuracy when the number of training samples is on the order of the number of

species. They then apply these methods to a panel of in vitro colonization experiments, in which two focal species (*E. faecalis* and *A. muciniphila*) are inoculated into ~300 synthetic communities derived from ~24 human fecal samples. They find that this sample size is sufficient to produce predictions for the probability of invasion and equilibrium abundance that are significantly better than chance, and they use these predictions to identify species with the largest impact on the colonization outcomes of the focal species.

Overall, I think this is a nice study, which significantly advances our understanding of the mechanisms of colonization resistance in microbial communities. I think it would be an excellent fit for Nature Communications. That being said, I think there are several places where the claims could be toned down a bit, to better reflect the reality of what has been shown. I also have some questions related to the details of the experimental methods and analysis. I describe these and other comments and suggestions in more detail below.

(1) Throughout the work, the authors use Receiver Operator Characteristic (ROC) curves to evaluate the accuracy of their colonization classifiers, and conclude that classification is accurate when the area under the curve ROC curve (AUROC) is greater than 0.8 (line 155).

While the use of these methods is standard in machine learning, I worry that using unqualified terms like "accurately predict" / "successfully predict" / "predictable" as a stand in for this technical definition might be confusing for microbial ecologists, who might be more inclined to interpret them according to their standard English usage.

What the manuscript strongly shows is that the trained classifiers are better than random guessing (AUROC=0.5), but there is a large gap between that and "successful prediction". I think the manuscript would be improved if the authors could revise their language throughout the work to make their accuracy limitations more clear. E.g., a plain reading of the current version of the abstract might give the impression that there are no accuracy limitations at all. The use of "predictable" in the section heading on line 196 is similarly confusing, as is the use of "can solve the classification problem" on line 233. The results are already impressive, so some extra clarity on this front would be helpful for avoiding confusion within the field.

(2) In addition to clarifying the language, it would be helpful to provide some guidance for how ROC curves and AUROC scores translate to accuracy in real experiments. E.g. what is the best total accuracy (true positives + true negatives / total) that can be achieved for a given classifier, regardless of its internal threshold? (this would work well for balanced examples like the one here, but would need to be tweaked for less permissive colonization)

(3) Likewise, to get some intuition for the AUC scores / accuracy scores, it could be useful to compare the machine learning methods to an even simpler classifier based on total community diversity. Given the results of Figs 3B and 4B, this naïve classifier might do pretty well...

(4) For the quantitative prediction task (predicting the relative abundance at steady state), the authors use the Pearson correlation as a measure of accuracy. I was confused the correlation metric was used, rather than raw sum of squared differences between the observed and predicted values. (The latter would seem to more closely match what one would be looking for in an experiment.)

Also, I couldn't tell from the methods whether the R^2 values were calculated from the relative abundances or their log transformed values; Fig 4H-J suggests the latter but would be good to clarify.

(5) The relative abundance prediction results did not appear to be shown for *E. faecalis* (but I assume that they were constructed in order to calculate the CI scores in Fig. 5). Regardless, I think it would be good to show the results of this prediction task for both species; if the results are not as good as they were for *A. muciniphila*, it would be good for readers to know that this prediction method is less accurate for some species.

(6) The introduction seems to set it up as a surprise that $O(N)$ training samples – rather than

$O(N^2)$ – are sufficient to enable accurate predictions. I didn't quite get this reasoning. The prediction is being done on one invading species at a time, so $O(N)$ training samples would seem correspond to the brute force expectation from GLV-like models (where the invasion fitness is a linear function of the current abundances). $O(N^2)$ would only be required if you wanted to repeat the colonization predictions for all possible species, which was not the problem considered here. I think it would be good for the authors to clarify this point, since the methods become much simpler to understand once one realizes that they are essentially following the brute force approach.

(7) I couldn't quite tell from the methods how the antibiotic-treated communities were derived. Was it a single pulse of antibiotics? Or continuous addition to the media (including during the invasion event)?

(8) I couldn't tell from the methods how the perturbations used to define "colonization impact" (CI) scores were chosen. What were the magnitudes and/or signs of the perturbations?

(9) It is known that GLV models exhibit limit cycles and chaotic dynamics, so that they do not always have a steady state. How would you apply the method in that case? Were the parameters chosen in a special way to avoid this behavior?

(10) I couldn't tell how the self-interaction terms were chosen in the GLV model. Please clarify.

(11) The GLV model works with absolute abundances, while the data are compositional (relative abundance). Could this influence the differences between the colonization interactions in Fig. 5B? Ideally, the authors could convert their GLV abundances into relative abundances to make their simulations more like their data. Is the analytical result in Eq. 4 still valid in the compositional domain?

(12) Similar results to Eq. 4 have been utilized in cavity method approaches to GLV models (e.g. Bunin Phys Rev E 2017). These should be cited around line 158-160, along with the new derivation.

(13) If I understand correctly, the result in follows almost immediately from the definition of the GLV model (similar to Eq. 2), if we make the additional assumption that A is invertible. This seems like a pretty strong assumption, so it might be best to say that *some* GLV models satisfy this derivation. Are the parameter chosen in a special way here to ensure that A is invertible?

(14) The use of "established" on line 174 does not seem appropriate, given that similar experiments were performed in the cited reference. I'd suggest changing this to something like "used" instead.

(15) The use of the term "the data driven approach" is confusing, since there are many possible ways to use data to inform prediction. I'd suggesting changing this to "data driven approaches" or "our data-driven approach", depending on the context, so that readers will know what is being referred to.

(16) Related to the previous point, several earlier studies have used data-driven approaches to predict competition and colonization gut microbiome-like communities (e.g. Buffie et al Nature 2015, Venturelli et al Mol Sys Biol 2018). They used slightly more parameterized models (e.g. GLV in this case) but are still conceptually related to the approaches present work. These contributions should be cited somewhere in the introduction (e.g. around line 80).

Response to Reviewer #1:

Point 1.0: In their work, Wu et al. investigated how easily *E. faecium* and *A. muciniphila* could invade 300 stool communities derived from 24 healthy donors supplemented with 12 different antibiotics. They saw that invasion success depended on stool source, antibiotic treatment, community diversity and biomass. They then predicted invasion success with three supervised classification approaches and found that *E. faecalis* had a large negative impact on the colonisation of *E. faecium*. Finally, they confirmed the relationship between *E. faecium* and *E. faecalis* experimentally. This is an important work, but I see a number of major issues.

Response: We thank the reviewer for acknowledging the importance of our work. Below we provide a point-by-point response to the reviewer's comments.

Major comments

Point 1.1: The stool communities are treated with 12 different antibiotics. Thus, the authors cannot describe communities as permissive or resistant since it is not clear whether the community composition per se or the antibiotic treatment hampered colonisation. In this context, we need to know how sensitive *E. faecium* and *A. muciniphila* are to the 12 antibiotics tested. Perhaps this sensitivity is as good as or even a better predictor of invasion success than the composition of the target communities. If this turns out to be the case, the entire point of this study is undermined and the authors would need to demonstrate that machine learning approaches can predict invasion success on untreated stool samples.

Response: Firstly, we apologize for not explaining our experimental details clearly, which caused the confusion raised by the reviewer. We did not add antibiotics as a continuous treatment. The stool-derived communities were treated with antibiotics for 24h on Day 0 (i.e. a single pulse). Afterwards, the communities were passaged five times (from Day 1 to Day 6) in antibiotic-free medium to reach a stable state before introducing the exogenous species. These details were originally provided in Materials lines 516-518: "After 24h of antibiotics treatment, *in vitro* microbial communities were passaged every 24 h with a 1:200 dilution into fresh medium". We have revised Figure 2 and the relevant paragraph in the main text to describe the experimental procedure clearly (lines 188-194, page 7).

"To increase the diversity in baseline communities, we treated each donor's sample with a single pulse of 12 antibiotics from different classes (Table S1). After 24 h of antibiotics treatment, in vitro microbial communities were passaged every 24 h with a 1:200 dilution into fresh medium (Fig.2A)."

Figure 2A. Large-scale invasion experiments in human stool-derived *in vitro* microbial communities. Schematic representation of *in vitro* culture of human stool-derived microbial communities in 96-well plates. Stool samples from 24 donors were treated with 12 different antibiotics for 24 h. The control group was not treated with antibiotics. All communities were passaged five times to reach a stable state, i.e., baseline community profiles (green dot).

Secondly, we performed a sanity-check experiment to make sure that any remaining antibiotics after serial dilutions ($>10^{11}$ fold dilution of the antibiotic concentration added on Day 0) does not influence the growth of the invading species *E. faecium* (**Fig.R1**).

Figure R1. A single pulse of antibiotics after serial dilutions does not influence the growth of *E. faecium*. The growth curve of *E. faecium* was measured in the presence of serially diluted antibiotics (1:200 dilution for 5 times). The initial concentration of 12 antibiotics is the same as listed in Table S1. OD_{600} was taken every 15 minutes over a 12-hour period, with each sample having three replicate wells. Fisher's The Least Significant Difference test with Bonferroni Adjustment was used to decide if there were any significant differences in growth rate between treatments. The maximum growth rate r (milli Optical Density per hour) with its standard error was estimated from microbial growth curves.

Taken together, we want to clarify that a single pulse of antibiotics was used on Day 0 to generate diverse baseline communities and afterwards no antibiotics were added to the medium. The antibiotics in our experiment shall not have any direct influence on the growth of the invading species (introduced on Day 6). Finally, we want to point out substantial variations in the post-invasion abundance of the invasive species under the same antibiotic treatments (**Fig.S11** for *E. faecium* and **Fig.S17** for *A. muciniphila*). These variations in colonization outcomes are clearly attributed to the variations in community composition (**Fig.S4**).

Point 1.2: The authors predict *E. faecium*/*E. faecalis* as a strongly interacting pair based on the importance of *E. faecalis* in Random Forest classification. Do the other two classification approaches also point to *E. faecalis*? This could be explored e.g. with LIME (<https://arxiv.org/abs/1602.04938>). There may be other strongly interacting species that were either not detected by the machine learning approaches or that were hidden by the effect of the antibiotic.

Response: We found that *Enterococcus faecalis*, as well as other 5 species among the top 20 list (including *Faecalibacterium prausnitzii*, *Ruminococcus gnavus*, *Blautia*_sp., *Clostridium*_sp. L.2.50, and *Roseburia inulinivorans*) were predicted by both Random Forest (**Fig. 5E**) and NODE (**Fig.S19B**) to have a strong impact on *E. faecium* colonization. In contrast, Logistic Regression

classifier did not identify *E. faecalis* as a strong inhibitor (**Fig.S19D**), and its predictive performance is substantially worse than Random Forest and NODE (**Fig.3**).

Secondly, following the reviewer’s suggestion, we applied LIME to explain the prediction of the random forest classifier (**Fig.19E**). The LIME analysis indicates that the presence of *E. faecalis* increases the probability that a community is resistant to *E. faecium* invasion. Other top features identified for resistant communities, including *Faecalibacterium prausnitzii*, *Clostridium perfringens*, and *Roseburia inulinivorans*, are consistent with the colonization impact prediction (**Fig.5E**).

We have also included this analysis in the revised manuscript (see lines 328-332, pages 10-11):
“The colonization impact of those top-ranking species is also consistent with the result of LIME which is a novel explanation technique that explains the predictions of any classifier (54), e.g., the presence of *E. faecalis*, *F. prausnitzii* and *C. perfringens* in resistant communities increases the probability that *E. faecium* cannot colonize (see **Fig.S19E**).”

Figure 5. Colonization impact in simulated and experimental communities. (D-E) The distribution of colonization impact on *E. faecium*, and the top-ranking species with negative colonization impact (median across different communities, RF classifier).

Fig. S19. Colonization impact in experimental communities predicted by NODE and logistic regression. (A-B) The distribution of colonization impact on *E. faecium*, and the top-ranking species with negative colonization impact (median across different communities) predicted by NODE. **(C-D)** The distribution of colonization impact on *E. faecium*, and the top-ranking species with negative colonization impact (median across different communities) predicted by logistic regression. **(E)** The top features of the Random Forest classifier identified by LIME. We applied LIME to interpret the random forest prediction for *E. faecium* colonization, with 80% as the training set and the remaining 20% as the test set. For each test sample, we selected the top 5 features and showed the mean and standard deviation of each feature's contribution among all test samples.

Point 1.3: L. 227-228: 240 training versus 60 test samples from 20 versus 4 donors is a very unbalanced design with a risk of overfitting. Does classification still work when training and testing are more balanced?

Response: Following the reviewer's suggestion, we evaluated the performance of machine learning approaches under a balanced split of training samples and test samples, i.e. 50% training samples versus the remaining 50% test samples. Overall, we found that the performance of machine learning classifiers with 50-50 split (**Fig.S13**) is similar to that of 80-20 split. The slight decrease in AUROC of machine learning classifiers can be attributed to the reduction in training sample size (**Fig.1**). We have included these results in the revised manuscript (lines 248-251, page 8):

“We also evaluated the performance of machine learning classifiers with a balanced split of training and test samples, showing that Random Forest remains the best classifier (AUROC=0.81 for *E. faecium* colonization, **Fig.S13A**).

Fig. S13. The colonization outcome prediction of *E. faecium* and *A. muciniphila* in human stool-derived *in vitro* microbial communities with balanced training-test split. (A-B) ROC curve of machine learning models in binary classification (permissive vs. resistant) of the colonization outcomes of *E. faecium* (A) and *A. muciniphila* (B).

Point 1.4: L. 338-340: "We demonstrate that the data-driven approach enables accurate function prediction and system-level understanding of complex microbial communities." This is a big claim. Which systems-level understanding did the authors gain? They did not learn the mechanism behind the interaction between *E. faecalis* and *E. faecium*. The advantage of knowing mechanisms is that the model is better able to deal with changes in conditions, such as lower pH or different nutrient levels. A machine learning approach may work in a given situation but will have to be retrained when circumstances, such as the medium, change. This weakness of the

data-driven approach is not discussed.

Response: We fully agree with the reviewer that the data-driven approach has its own weaknesses. We have removed “system-level understanding” from the original sentence (line 338, page 11) and added discussion on the limitation of the data-driven approach (lines 434-436, page 13):

“Lastly, while the data-driven framework can be generalized to different scenarios, the machine learning model must be re-trained when the environmental condition changes. In contrast, mechanism-based models can better deal with changes in conditions (e.g., pH, nutrient level).”

Point 1.5: The gLV simulations deviate from the real case in an important point: the exogenous species may have been already present in the stool communities, perhaps below detection level. In that case, not the colonisation itself is of interest but the boost given to the “exogenous” species (which may be due to some density-dependent mechanism). This issue needs to be discussed.

Response: We thank Reviewer #1 for this critical comment. We have added the discussion on this point in the revised main text (lines 438-443, pages 13-14):

“We noted a potential difference between the gLV simulations and experimental data: the exogenous species may already be present in the stool-derived communities, but its steady-state abundance was below the detection threshold. In this scenario, the introduction of the exogenous species (~5% of community biomass) may provide a growth boost (e.g., via some density-dependent mechanism) and enable the species to co-exist with other species at a higher steady-state abundance (i.e., multiple stable states).”

Point 1.6: Another problem is that no noise was added to the simulated data. However, sequencing comes with large technical variation, so it is important to assess the robustness of the machine learning tools to such variation.

Response: Following the reviewer’s suggestion, we added varying levels of noise in the simulated data to assess the robustness of machine learning models. We found that the prediction performance for classification is robust against noise (**Fig.S3A**), while the prediction performance for regression had a slight decrease for strong noise $\epsilon \sim 0.1$ (**Fig.S3B**). We have included these results in the revised manuscript (lines 178-181, page 6):

*“Finally, we added varying levels of noise in the simulated data to assess the robustness of machine learning models to technical variations (e.g., measurement errors). For both the classification problem and the regression problem, we found that the predictive performance of machine learning models is robust against noise (**Fig.S3**).”*

Fig. S3. Prediction of colonization outcomes in simulated data with noise. We first generated noise drawn from the normal distribution with mean 0 and standard deviation ε . Then, we added the noise into microbial composition: $\tilde{p} = \max(p(1 + \mathcal{N}(0, \varepsilon)), 0)$, with ε representing the noise strength. We used three ε values corresponding to no noise ($\varepsilon = 0$), weak noise ($\varepsilon = 0.01$), and strong noise ($\varepsilon = 0.1$), respectively. **(A)** Evaluation of the data-driven approach in solving the classification task of COP. AUROC of three machine learning models, including Logistic Regression (LR), COP-Neural Ordinary Differential Equations classifier (NODE), and Random Forest classifier (RF). **(B)** Evaluation of the data-driven approach in solving the regression task of COP. Pearson correlation between the true abundance and the abundance predicted by three machine learning models, including Elastic Net Linear Regression (ENET), COP-NODE regressor (NODE), and Random Forest regressor (RF). The models were trained with 200 samples, and network connectivity $C = 0.3$.

Point 1.7: L. 165-166 and elsewhere: when assessing prediction accuracy for the abundance of a species, the Pearson correlation is not ideal, since it does not consider abundances themselves, only trends. It is more informative to report the average ratio between predicted and observed abundance here. When comparing predicted and observed community compositions (e.g. Fig S7), Pearson is also not appropriate and better replaced by Bray Curtis.

Response: Following the reviewer’s suggestion, we reported the ratio between the predicted abundance and the observed abundance in the revised manuscript (lines 170-172, page 6):

“The predictive performance was evaluated with Pearson’s correlation coefficient between the predicted and true abundance (log-transformed), as well as the ratio between the predicted abundance and the true abundance (Fig.S2)”.

Fig. S2. The ratio between the predicted abundance and the true abundance. The parameters and labels are identical to Figure 1, panels F-H.

Secondly, following the reviewer’s suggestion, we used Bray-Curtis dissimilarity to compare community compositions and included the results in **Fig.S9**. We want to point out that there is a typo in the reviewer’s comment. In **Fig.S9**, we are comparing the community compositional profiles between replicates, not between predicted and observed profiles.

Fig. S9. The composition of in vitro communities before and post *E. faecium* invasion is highly reproducible across replicates. (C-D) Distribution of Bray-Curtis dissimilarity between the community compositional profiles of replicates before (C) and post (D) *E. faecium* invasion.

Point 1.8: Is the OD correlated to actual measurements of biomass (gram dry weight) or to cell densities derived from flow cytometry?

Response: In light of the reviewer’s comment, we performed experiments to examine the correlation between OD₆₀₀ measurements and the cell density of six different stool-derived communities (Fig.S24). We found that OD₆₀₀ is highly correlated to cell densities derived from flow cytometry, thus lending support to using OD₆₀₀ to estimate the absolute abundance of stool-derived communities in our study.

We have added the experimental details in the revised Supplementary Methods (lines 115-127, pages 4-5):

“Estimating the absolute abundance of stool-derived microbial communities through optical density (OD₆₀₀) Measurements. To verify whether OD₆₀₀ is an effective measure for estimating microbial total biomass, we conducted experiments to assess the correlation between OD₆₀₀ readings and cell density in stool-derived communities (see Fig.S24). Six different stool-derived microbial communities were thawed and revived by adding 20 μL of the stocks to 980 μL of BHI medium in deep well plates. After incubation for 24 hours at 37°C, the saturated cultures were diluted into 1 mL of BHI in a new plate at five different dilution ratios (1:1.5, 1:2, 1:4, 1:8 and 1:16). Subsequently, the OD₆₀₀ of these diluted communities was measured using an Epoch 2 plate reader (BioTek). Parallely, we assessed the particle density of these communities using CytoFLEX s cytometry (Beckman Coulter). Our findings reveal a strong correlation between OD₆₀₀ values and cell densities determined via flow cytometry, supporting the use of OD₆₀₀ as a reliable method for estimating the absolute abundance of stool-derived microbial communities in our study.”

Fig. S24. Optical density measurement (OD₆₀₀) is highly correlated with cell density derived by flow cytometry. Data were collected using six different stool-derived communities. The error bars indicate the standard error for each community (n=3 replicates).

Minor comments

Point 1.9: How variable are stool communities coming from the same donor and with the same treatment? Is that variability exceeded by the variability across treatments (per donor)?

Response: In our experimental system, the variability of stool-derived communities across replicates (see **Fig S9C** above) is exceeded by the variability across subjects/donors, and then exceeded by the variability across treatments (**Fig S5D**).

Fig. S5. Generation of diverse baseline communities by antibiotics treatments. (C) The colored dot represents the compositional profile of each subject averaged over 12 antibiotic treatments. Error bars are SEMs. (D) The density plot illustrating Bray-Curtis dissimilarity in compositional profiles at the species level across various subjects and treatments.

Point 1.10: Why do detection thresholds for *E. faecium* and *A. muciniphila* differ by an order of magnitude?

Response: We thank Reviewer #1 for this very critical comment. In this work, we determined the detection thresholds by a spike-in experiment (**Fig. S23 and Methods**). The detection thresholds for different microbial species can indeed vary significantly, often by an order of magnitude, due to several factors: 1) Genetic and Morphological Differences: The genetic and morphological characteristics of microbes can affect detection. For instance, certain bacteria might have unique cell wall structures that make them more difficult to lyse, or their DNA may be less efficiently amplified in PCR assays. 2) Background Microbial Flora: The presence of a complex background microbial community can obscure the detection of certain species, especially those present in lower abundances.

Point 1.11: Another limitation to be discussed is the presence of mucin in liquid form. I assume this was necessary to enable the growth of *A. muciniphila*, but it deviates from the biology of the intestinal system, where *A. muciniphila* prefers to colonise (solid) mucin and other gut bacteria prefer the lumen. It would be of interest to find out to what extent *A. muciniphila*'s ability to colonise the tested communities depends on the mucin concentration in the medium.

Response: We thank the reviewer for this comment. In the revised manuscript, we have included the following discussion on the role of mucin (lines 429-433, page 13):

“Fourth, in our in vitro experiments, we found that A. muciniphila was able to stably colonize in the majority of stool-derived communities with relatively high abundance. It is known that mucin is the preferred nutrient source of A. muciniphila, so it would be interesting to study to which degree the colonization of A. muciniphila depends on the mucin concentration provided in the medium.”

Finally, we thank Reviewer #1 again for reviewing our manuscript and providing the constructive comments that help us improve our work. We hope that our responses have addressed all the comments in a satisfactory manner.

Response to Reviewer #2:

Point 2.0: This is an exciting paper using machine-learning approaches to predict the ability of a species to invade an existing community, as well as the final abundance of the invading species. Then they evaluate the same methods with simulated data (using a generalized Lotka-Volterra model) and with real communities using replicated self-assembly communities and antibiotics to create variation in community contexts. This paper is an important contribution to the field. My main concern is that in the description of the results, some negative or less interesting results are not described. I think the paper is really good but omitting or glossing over negative results can generate distrust in the reader. In particular: I have a few large comments that are more discussion points than criticisms of the paper:

Response: We thank the reviewer for acknowledging the importance of our work. Below we provide a point-by-point response to the reviewer's comments.

Point 2.1: To what extent maybe simpler statistical models (including LASSO regression) still predict the data? Are there less data-intensive methods that could still provide accurate predictions?

Response: For synthetic data generated by the classical Generalized Lotka–Volterra (GLV) model in community ecology, we did see that the simple models, e.g., linear regression and Elastic Net Linear Regression work well for both classification problems (**Fig.1C-E**) and regression problems (see **Fig.1F-H**). However, those simple statistical models did not work well in predicting the colonization of *A. muciniphila* (see **Fig.4H**). We anticipate that real microbial communities are way more complicated than the simple GLV model (which only includes pair-wise inter-species interactions). Sophisticated machine learning models may have to be leveraged to predict colonization outcomes for complex communities. We have included this discussion in the revised manuscript (lines 401-408, page 13).

“We anticipate that real microbial communities are way more complicated than the simple GLV model (which only includes pair-wise inter-species interactions). Sophisticated machine learning models may have to be leveraged to predict colonization outcomes for complex communities. We anticipate that more training samples are required if high-order interactions are considered. However, those high-order interactions might be weak and do not significantly impact the prediction. However, their impact on the prediction could be weak as the community-function landscapes display a low degree of ruggedness (70).”

Point 2.2: Why does random-forest performs poorly in the regression task with the Lotka-Volterra simulations but it is the best-performing algorithm in real communities? Any hypotheses?

Response: It is true that there is a discrepancy between the performance of Random Forest in gLV simulated data and real data. This could be due to 1) the dynamics of the gLV model may be different from that of experimental communities. For instance, when the gLV model has globally stable equilibria, the final state is solely determined by the species collection. 2) the distribution of interspecies interaction strength used in the gLV model may differ from experimental communities. In experimental communities, a few strongly interacting species may dominate the contribution to the colonization resistance of exogenous species, while in simulated data the contribution is more evenly distributed among resident species. We have included this discussion in the revised manuscript (lines 443-451, page 14):

“Moreover, there is a discrepancy between the performance of Random Forest in gLV simulated data and real data. Potential explanations includes: 1) the dynamics of the gLV model may be different from that of experimental communities. For instance, when the gLV model has globally stable equilibria, the final state is solely determined by the species collection. 2) the distribution

of interspecies interaction strength used in the gLV model may be different from experimental communities. In experimental communities, a few strongly interacting species may dominate the contribution to the colonization resistance of exogenous species. In contrast, in simulated data, the contribution is more evenly distributed among resident species.”

Point 2.3: Emphasize in the discussion that these predictions require communities in a steady state (or to what extent are the methods robust to this assumption).

Response: We have revised the discussion to emphasize this point (lines 370-373, page 12):
“In comparison, our approach is based on sampling an ensemble of complex communities (~100 species, Fig.S4) and using the sampled communities to infer the mapping between community composition and colonization outcomes (61) by assuming the compositional profiles represent steady states of the local communities.”

Smaller comments:

Point 2.4: L100- I wonder if there is a way to map these variables in figure 1 for extra clarity.

Response: We thank the reviewer for the suggestion. We have revised **Figure 1** to illustrate the variables (N: species in the meta-community; M: number of local communities sampled)

Figure 1A. Prediction of colonization outcomes for complex microbial communities via the data-driven approach.

Point 2.5: L170 - State in the Lotka-Volterra results that random forest performs poorly at all sizes.

Response: We have revised the manuscript accordingly (lines 175-178, page 6):
“For training sample size $S_{train} = 2N$ or higher, there was a substantial improvement in the quantitative prediction of the post-invasion abundance by Elastic Net Linear Regression and NODE; in contrast, Random Forest had a poor performance at all sample sizes.”

Point 2.6: L230. Why are regression results not presented for *E. faecium*? in the main text and figure 3?

Response: For regression, we need training samples with non-zero post-invasion abundance (i.e., permissive communities). Because *E. faecium* only colonized in ~30% of communities in our experiments, we do not have sufficient samples to train the regression models ($S_{train} < N$). We have clarified this point in the revised manuscript (lines 237-240, page 8):

*“For the regression problem, we need training samples with non-zero post-invasion abundance. Because *E. faecium* only colonized in ~30% of baseline communities in our experiments, the number of samples is insufficient to train the regression models to predict the post-invasion absolute abundance.”*

Point 2.7: Figure 1. Make invading species clearer, maybe the lines of other species in the community should be shades of gray.

Response: We have revised **Figure 1** following the reviewer's suggestions (see **Fig.1A** above).

Point 2.8: Figure 2B. Is this real data? Which communities are these?

Response: **Figure 2B** is not real data but a schematic representation which was designed to illustrate an example of in vitro baseline communities with diverse composition. This has been specified in the updated figure legend for clarity.

Point 2.9: Figure 6 - Having the legend on each side of the dashed line is confusing, makes it seem like one strain dominates on one side of the line vs. the other (until carefully reading the caption).

Response: We thank the reviewer for pointing out the confusion. We have moved the position of legends in **Figure 6**.

Finally, we thank Reviewer #2 again for reviewing our manuscript and providing the constructive comments that help us improve our work. We hope that our responses have addressed all the comments in a satisfactory manner.

Response to Reviewer #3:

Point 3.0: In this manuscript, Wu & Wang et al study the problem of predicting whether a given bacterial species will successfully invade a larger microbial community. They utilize a “data driven” approach, in which the observed colonization outcomes of a given invader across a large number of “training” communities are used as input data to fit machine learning models to extrapolate colonization outcomes to other “test” communities. They demonstrate this approach on simulated data from Generalized Lotka-Volterra models, and show that several machine learning models have reasonable accuracy when the number of training samples is on the order of the number of species. They then apply these methods to a panel of in vitro colonization experiments, in which two focal species (*E. faecalis* and *A. muciniphila*) are inoculated into ~300 synthetic communities derived from ~24 human fecal samples. They find that this sample size is sufficient to produce predictions for the probability of invasion and equilibrium abundance that are significantly better than chance, and they use these predictions to identify species with the largest impact on the colonization outcomes of the focal species. Overall, I think this is a nice study, which significantly advances our understanding of the mechanisms of colonization resistance in microbial communities. I think it would be an excellent fit for Nature Communications. That being said, I think there are several places where the claims could be toned down a bit, to better reflect the reality of what has been shown. I also have some questions related to the details of the experimental methods and analysis. I describe these and other comments and suggestions in more detail below.

Response: We thank the reviewer for acknowledging the importance of our work. Below we provide a point-by-point response to the reviewer’s comments.

Point 3.1: Throughout the work, the authors use Receiver Operator Characteristic (ROC) curves to evaluate the accuracy of their colonization classifiers, and conclude that classification is accurate when the area under the curve ROC curve (AUROC) is greater than 0.8 (line 155). While the use of these methods is standard in machine learning, I worry that using unqualified terms like “accurately predict” / “successfully predict” / “predictable” as a stand in for this technical definition might be confusing for microbial ecologists, who might be more inclined to interpret them according to their standard English usage. What the manuscript strongly shows is that the trained classifiers are better than random guessing (AUROC=0.5), but there is a large gap between that and “successful prediction”. I think the manuscript would be improved if the authors could revise their language throughout the work to make their accuracy limitations more clear. E.g., a plain reading of the current version of the abstract might give the impression that there are no accuracy limitations at all. The use of “predictable” in the section heading on line 196 is similarly confusing, as is the use of “can solve the classification problem” on line 233 . The results are already impressive, so some extra clarity on this front would be helpful for avoiding confusion within the field.

Response: We thank Reviewer #3 for this very constructive comment. To avoid potential confusion within the field of microbial ecology, we have revised our language throughout the manuscript to better describe prediction performance. In particular, we have removed unqualified terms, e.g., “accurately predict”, “successfully predict”, “predictable”. Instead, we used the language “with high prediction performance (in terms of AUROC>0.8)”. Moreover, in the manuscript, we explicitly mentioned that a perfect classifier (prediction model) has AUROC=1, while for random guess AUROC=0.5. This will help readers better interpret our prediction performance (see line 154, page 6):

“For network connectivity $C = 0.3$, we found that the Area Under the Receiver Operating Characteristic curve (AUROC, a perfect classifier has AUROC=1 and AUROC=0.5 for random guess) of three machine learning models was above 0.9 with training sample size $S_{train} = N$.”

Point 3.2: In addition to clarifying the language, it would be helpful to provide some guidance for how ROC curves and AUROC scores translate to accuracy in real experiments. E.g. what is the best total accuracy (true positives + true negatives / total) that can be achieved for a given classifier, regardless of its internal threshold? (this would work well for balanced examples like the one here, but would need to be tweaked for less permissive colonization)

Response: In the revised manuscript, we computed the accuracy of each classifier in the simulated data and the real dataset and included both AUROC and accuracy scores in relevant figures (see **Fig.3-4**). Specifically, for *E. faecium*, the AUROC score is 0.71 for LR, 0.81 for NODE, and 0.86 for RF; the Accuracy is 0.68 for LR, 0.75 for NODE, and 0.82 for RF; For *A. muciniphila*, the AUROC score is 0.75 for LR, 0.79 for NODE and 0.84 for RF; the Accuracy is 0.51 for LR, 0.73 for NODE and 0.78 for RF. We have added this result in the revised manuscript (lines 245-248, page 8):

“Random Forest classifier displayed the best performance in predicting whether E. faecium could successfully colonize based on the species-level community composition (AUROC=0.86, Accuracy=0.82), followed by COP-NODE classifier (AUROC=0.81, Accuracy=0.81) and Logistic Regression (AUROC=0.71, Accuracy=0.75).”

Point 3.3: Likewise, to get some intuition for the AUC scores / accuracy scores, it could be useful to compare the machine learning methods to an even simpler classifier based on total community diversity. Given the results of Figs 3B and 4B, this naïve classifier might do pretty well...

Response: Following the reviewer’s suggestion, we have performed the analysis on using the community diversity (i.e. species richness) for classification (see **Fig.S14**). For colonization outcomes of *E. faecium*, the AUROC score based on species richness is 0.78, while the AUROC score of Random Forest classifier is 0.86 (**Fig.3G**). For colonization outcomes of *A. muciniphila*, the AUROC score based on species richness is 0.74, while the AUROC score of Random Forest classifier is 0.84 (**Fig. 4G**). The results indicated that species richness alone can be used as a simple classifier, but its prediction performance is worse than machine learning methods based on the taxonomic profile. Our analysis is consistent with the observation of a significantly negative correlation between community diversity and the post-invasion steady state abundance of the invading species. Nevertheless, we note that the relation between community diversity and colonization resistance is system-specific and determined by the interaction strength between the resident species and the invading species (see **Supplementary Text**).

We have included this analysis in the revised manuscript (lines 251-254, page 8).

*“For comparison, we performed the analysis on using the community diversity (i.e. species richness) for classification (see **Fig.S14**). The results indicated that species richness alone can be used as a simple classifier, but its prediction performance is worse than machine learning methods based on the taxonomic profile.”*

Fig. S14. The colonization outcome of *E. faecium* and *A. muciniphila* in human stool-derived *in vitro* microbial communities using species richness as the predictor. ROC curve of machine learning models in binary classification (permissive vs. resistant) of the colonization outcomes of *E. faecium* (A) and *A. muciniphila* (B) using richness as predictors. For each 6-fold cross-validation (ROC curves shown in a light color), we used the species richness as the colonization probability to compute AUROC.

Point 3.4: For the quantitative prediction task (predicting the relative abundance at steady state), the authors use the Pearson correlation as a measure of accuracy. I was confused the correlation metric was used, rather than raw sum of squared differences between the observed and predicted values. (The latter would seem to more closely match what one would be looking for in an experiment.)

Response: We computed the mean squared differences between the observed and predicted values and included them in **Fig.4H-J** and **Fig.S18**.

Point 3.5: Also, I couldn't tell from the methods whether the R^2 values were calculated from the relative abundances or their log transformed values; Fig 4H-J suggests the latter but would be good to clarify.

Response: We used the log-transformed values to compute the Pearson's correlation. We have clarified this point in **Figure 4** legends.

Point 3.6: The relative abundance prediction results did not appear to be shown for *E. faecalis* (but I assume that they were constructed in order to calculate the CI scores in Fig. 5). Regardless, I think it would be good to show the results of this prediction task for both species; if the results are not as good as they were for *A. muciniphila*, it would be good for readers to know that this prediction method is less accurate for some species.

Response: For regression, we need training samples with non-zero post-invasion abundance (i.e., permissive communities). Because *E. faecium* only colonized in ~30% of communities in our experiments, we do not have sufficient samples to train the regression models ($S_{train} < N$). We have clarified this point in the revised manuscript (lines 237-240, page 8):

*“For the regression problem, we need training samples with non-zero post-invasion abundance. Because *E. faecium* only colonized in ~30% baseline communities in our experiments, the number of samples is insufficient to train the regression models to predict the post-invasion absolute abundance.”*

The Colonization Impact score of *E. faecalis* was computed by comparing the colonization probabilities of *E. faecalis* before and after perturbation (see Supplementary Materials lines 70-72, page 3):

“For classification models, x_i^α and \tilde{x}_i^α represents the colonization probability before and after perturbing the abundance of species i , respectively.”

Point 3.7 The introduction seems to set it up as a surprise that $O(N)$ training samples – rather than $O(N^2)$ – are sufficient to enable accurate predictions. I didn't quite get this reasoning. The prediction is being done on one invading species at a time, so $O(N)$ training samples would seem correspond to the brute force expectation from GLV-like models (where the invasion fitness is a linear function of the current abundances). $O(N^2)$ would only be required if you wanted to repeat the colonization predictions for all possible species, which was not the problem considered here. I think it would be good for the authors to clarify this point, since the methods become much simpler to understand once one realizes that they are essentially following the brute force approach.

Response: We agree with the reviewer that $O(N)$ training samples are sufficient to enable accurate prediction of invasion resistance, as shown by analytical derivation of gLV model and simulations. Our conclusion of $O(N)$ training samples was drawn from the analytical solution of gLV dynamics with pairwise interactions. We anticipate that more training samples are required if high-order interactions are considered. However, those high-order interactions might be weak and do not significantly impact the prediction. This has also been demonstrated in a recent study that community-function landscapes display a low degree of ruggedness and are dominated by low-order terms (1).

In the revised manuscript, we have removed $O(N^2)$ from the introduction (line 68, page 3) and expanded the relevant discussion on sample size requirement (lines 404-408, page 13):

“We anticipate that more training samples are required if high-order interactions are considered. However, those high-order interactions might be weak and do not significantly impact the prediction. However, their impact on the prediction could be weak as the community-function landscapes display a low degree of ruggedness (70).”

Point 3.8: I couldn't quite tell from the methods how the antibiotic-treated communities were derived. Was it a single pulse of antibiotics? Or continuous addition to the media (including during the invasion event)?

Response: Firstly, we apologize for not explaining our experimental details clearly, which caused the confusion raised by the reviewer. We did not add antibiotics as a continuous treatment. The stool-derived communities were treated with antibiotics for 24h on Day 0 (i.e. a single pulse). Afterwards, the communities were passaged five times (from Day 1 to Day 6) in antibiotic-free medium to reach a stable state before the introduction of the exogenous species. These details were originally provided in Materials Line 516-518: “After 24h of antibiotics treatment, *in vitro* microbial communities were passaged every 24 h with a 1:200 dilution into fresh medium”. We have revised Figure 2 and the relevant paragraph in the main text to describe the experimental procedure clearly (lines 188-194, page 7).

*“To increase the diversity in baseline communities, we treated each donor's sample with a single pulse of 12 antibiotics from different classes (Table S1). After 24h of antibiotics treatment, *in vitro* microbial communities were passaged every 24 h with a 1:200 dilution into fresh medium (Fig.2A).”*

Point 3.9: I couldn't tell from the methods how the perturbations used to define “colonization impact” (CI) scores were chosen. What were the magnitudes and/or signs of the perturbations?

Response: We thank the reviewer for pointing this out, and we apologize for not mentioning this in the manuscript. For the perturbation to evaluate colonization impact, we increase the abundance of a certain resident species in the community. The magnitude of the increase is 0.007, which represents ~5% of the total biomass of the community. Then, we can compare the colonization abundances of invading species with and without the perturbation.

In the revised manuscript, we have added this information to the section on colonization impact (see Supplementary Materials lines 65-66, page 3):

“The perturbation is performed by increasing the abundance of a certain resident species by 0.007, which represents ~5% of the total biomass.”

Point 3.10: It is known that GLV models exhibit limit cycles and chaotic dynamics, so that they do not always have a steady state. How would you apply the method in that case? Were the parameters chosen in a special way to avoid this behavior?

Response: We thank Reviewer #3 for this critical comment. In our simulation, the diagonal element of the interaction matrix is set to $d = -1$, and non-diagonal elements are drawn from a normal distribution with mean 0 and standard deviation σ . The number of species $S = 100$, and the interaction strength $\sigma = 0.1$, and network connectivity $C = 0.3, 0.4, 0.5$ ensure that the system is stable, e.g., $\sigma\sqrt{SC} < -d$.

Point 3.11: I couldn't tell how the self-interaction terms were chosen in the GLV model. Please clarify.

Response: We apologize for not mentioning this. In the revised manuscript, we have added this detail (see Supplementary Materials line 22, page 2).

"The diagonal elements of A are set to be $a_{ii} = -1$ to ensure the stability of the system."

Point 3.12: The GLV model works with absolute abundances, while the data are compositional (relative abundance). Could this influence the differences between the colonization interactions in Fig. 5B? Ideally, the authors could convert their GLV abundances into relative abundances to make their simulations more like their data. Is the analytical result in Eq. 4 still valid in the compositional domain?

Response: We apologize for any confusion arising from our initial description of the abundance. It is important to clarify that the experimental data used for prediction in our study consists of absolute abundances ($OD_{600} \times$ relative abundance). Therefore, there is no need to convert the simulated data from the gLV model into compositional profiles for comparison with the experimental data. This information was originally provided in the Methods lines 584-586: "The colonization outcomes were evaluated based on the invader's absolute abundance in the community, which was estimated by multiplying the relative abundance and the OD_{600} value ($OD_{600} \times$ relative abundance)." In the revised manuscript, we have included this information in the revised manuscript. Furthermore, it is important to note that the analytical result in Eq.4 is not valid for compositional profiles and should be interpreted within the context of absolute abundances.

Point 3.13: Similar results to Eq. 4 have been utilized in cavity method approaches to GLV models (e.g. Bunin Phys Rev E 2017). These should be cited around line 158-160, along with the new derivation.

Response: We thank Reviewer #3 for pointing this out. We have cited it in the revised manuscript (line 132).

Point 3.14: If I understand correctly, the result in follows almost immediately from the definition of the GLV model (similar to Eq. 2), if we make the additional assumption that A is invertible. This seems like a pretty strong assumption, so it might be best to say that *some* GLV models satisfy this derivation. Are the parameter chosen in a special way here to ensure that A is invertible?

Response: We thank Reviewer #3 for this critical comment. In the analytical analysis, we did assume that the interaction matrix A is invertible. However, in our numerical simulations, we did not choose parameters in any special way to force A to be invertible. Instead, we followed previous studies to set all the diagonal elements of A to be $a_{ii} = -1$, with certainly probability C (or $1-C$) set the off-diagonal elements a_{ij} to be a random number drawn from a normal distribution (or 0), respectively. We know that a square matrix is invertible if and only if zero is not an eigenvalue of this matrix. For our randomly constructed A matrix, the probability of having a zero eigenvalue is extremely low. Hence, we believe that the A matrix randomly constructed in our simulations is almost surely invertible.

We have revised the description in the revised manuscript (see lines 162-165, page 6):

“For the GLV model (with the interaction matrix A being invertible, which is almost surely true for randomly constructed matrices), our analytical derivations discovered a surprisingly simple linear relation between the post-invasion abundance of the exogenous species and the pre-invasion abundance of resident species.”

Point 3.15: The use of “established” on line 174 does not seem appropriate, given that similar experiments were performed in the cited reference. I’d suggest changing this to something like “used” instead.

Response: We have revised the description accordingly in the revised manuscript.

Point 3.16: The use of the term “the data driven approach” is confusing, since there are many possible ways to use data to inform prediction. I’d suggesting changing this to “data driven approaches” or “our data-driven approach”, depending on the context, so that readers will know what is being referred to.

Response: We have revised the description to “data driven approaches” in the revised manuscript.

Point 3.17: Related to the previous point, several earlier studies have used data-driven approaches to predict competition and colonization gut microbiome-like communities (e.g. Buffie et al Nature 2015, Venturelli et al Mol Sys Biol 2018). They used slightly more parameterized models (e.g. GLV in this case) but are still conceptually related to the approaches present work. These contributions should be cited somewhere in the introduction (e.g. around line 80).

Response: We have cited these studies in the revised manuscript (lines 54, 367).

Finally, we thank Reviewer #3 again for reviewing our manuscript and providing constructive comments that help us improve our work. We hope that our responses have addressed all the comments in a satisfactory manner.

References

1. A. Skwara *et al.*, Statistically learning the functional landscape of microbial communities. *Nat Ecol Evol*, (2023).

Reviewers' Comments:

Reviewer #1:

Remarks to the Author:

The authors have addressed all my concerns and I thus support the publication of this interesting study.

Reviewer #3:

Remarks to the Author:

This is my second review of the paper by Wu & Wang et al. Overall I think the manuscript is much improved – I particularly liked the addition of the new diversity-based predictor, which provides a useful calibration point for the more elaborate ML methods. However, I still have some concerns regarding the presentation of the results and prior literature that were not fully addressed by the revised version. I describe these and other issues in more detail below.

(1) In their response to my point 1 from my previous review, the authors mentioned that they removed unqualified terms like “accurately predict”, “successfully predict”, “predictable” in favor of more qualified versions. The changes in the abstract are great, but the section titles on lines 205 and 255 still seem to use the unqualified “can be predicted” language (and the titles of Figs 3 and 4 are only slightly better with the more vague use of “discriminable”).

As I mentioned in my last review, I think this binary language potentially oversells the present results, and that a more qualified statement like the abstract would be more appropriate. Alternatively, my concerns would also be addressed by switching from a declarative title to a more topic-oriented one like “Predicting the colonization outcomes of *A. muciphila*”, etc.

(2) I thank the authors for adding the accuracy values of their trained ML classifiers. To help interpret these numbers, it would help to compare them to the max accuracy of the naïve classifier (which would roughly correspond to the proportion of the most common outcome in the dataset).

(3) I also thank the authors for adding the new classifier based on community diversity (Fig. S14). However, I could not tell from the caption how this new classifier was constructed (line 299 just uses the generic phrase “machine learning models” – did they use the same 3 techniques they used for the species abundance predictors in Figs. 3 and 4? Please clarify.

(4) In addition, the description of Fig. S14 analysis in the main text (lines 249-250) leaves out the crucial quantitative information that it is only *slightly* worse than the more elaborate abundance-based models (a difference in AUC of <0.1, and equal or better than logistic regression results in Figs. 3 and 4). I think it would be helpful to list the AUROC (+ accuracy) of this diversity-based model in the main text, since it is critical for evaluating the advantage of the more models based on species abundance.

(5) In their response to point #13 from my last review, the authors added a new citation to the Bunin (2017) paper. However, the citation seems to have been added in a somewhat strange place (line 132, describing the setup of the GLV simulations) rather than the introduction of Eq. 4 (where it is most relevant). This omission, coupled with the “discovered” language used in the text, might give the misleading impression that Eq. 4 is a novel result.

(6) Likewise, in their response to point #17 from my previous review, the authors added citations to the Buffie et al 2015 and Venturelli et al 2018 papers in the introduction, but again in a rather oblique way (line 54) in a separate discussion that is distinct from the discussion of data driven approaches on line 80 where they are most relevant. Without acknowledging this prior work, I worry that the paragraph on lines 72-80 does not accurately situate the present study within the current state of the field. I think it is important to include some discussion of these existing approaches in the Introduction around line 80, even if it means that the Intro needs to be expanded somewhat (it is already quite short, so I think there should be plenty of space).

Minor:

(7) I thank the authors for including the details about how the antibiotics were added to the experiments (point #7 in my previous review). The description in the response letter is great, but the edits to the text could be modified slightly to indicate the length of passaging that was performed before colonization started. Also, it would be helpful to repeat or elaborate on this description in the methods section (e.g. around line 522) for readers who might overlook it within the main text.

(8) Line 94 claims that "Our results suggest that the function of complex microbial communities can be predicted via data-driven approaches and tunable." I don't really see how this claim is supported by the present study (since it doesn't directly look at community function or tunability). Please tone down or clarify.

(9) On line 85, it would be helpful to clarify that this is $O(N)$ colonization outcomes *per colonizing species* (otherwise, it runs the risk of getting confused with the $O(N)$ vs $O(N^2)$ issue).

Reviewer #1 (Remarks to the Author):

Point 1.0: The authors have addressed all my concerns and I thus support the publication of this interesting study.

Response: We thank Reviewer #1 for reviewing our manuscript again. We are glad to know that the reviewer has been satisfied with our previous response.

Reviewer #1 (Remarks on code availability):

Point 1.1: I did not review the code, but I had a look at the github, which provides a README but no installation instructions.

Response: We thank Reviewer #1 for this comment. We have added the installation instructions.

Reviewer #3 (Remarks to the Author):

Point 3.0: This is my second review of the paper by Wu & Wang et al. Overall I think the manuscript is much improved – I particularly liked the addition of the new diversity-based predictor, which provides a useful calibration point for the more elaborate ML methods. However, I still have some concerns regarding the presentation of the results and prior literature that were not fully addressed by the revised version. I describe these and other issues in more detail below.

Response: We thank Reviewer #3 for reviewing our manuscript again. We are glad to know that the reviewer has been satisfied with most of our previous responses. Below, we provide a point-by-point response to the reviewer's comments.

Point 3.1: In their response to my point 1 from my previous review, the authors mentioned that they removed unqualified terms like “accurately predict”, “successfully predict”, “predictable” in favor of more qualified versions. The changes in the abstract are great, but the section titles on lines 205 and 255 still seem to use the unqualified “can be predicted” language (and the titles of Figs 3 and 4 are only slightly better with the more vague use of “discriminable”).

As I mentioned in my last review, I think this binary language potentially oversells the present results, and that a more qualified statement like the abstract would be more appropriate. Alternatively, my concerns would also be addressed by switching from a declarative title to a more topic-oriented one like “Predicting the colonization outcomes of *A. muciniphila*”, etc.

Response: We thank Reviewer #3 for this comment, and we apologize for not addressing this comment thoroughly. In the revised manuscript, we have removed unqualified terms in section titles.

Point 3.2: I thank the authors for adding the accuracy values of their trained ML classifiers. To help interpret these numbers, it would help to compare them to the max accuracy of the naïve classifier (which would roughly correspond to the proportion of the most common outcome in the dataset).

Response: We thank Reviewer #3 for this comment. We have added the description of comparing with the accuracy value of the naïve classifier.

Point 3.3: I also thank the authors for adding the new classifier based on community diversity (Fig. S14). However, I could not tell from the caption how this new classifier was constructed (line 299 just uses the generic phrase “machine learning models” – did they use the same 3 techniques they used for the species abundance predictors in Figs. 3 and 4? Please clarify.

Response: We thank Reviewer #3 for pointing this out. We have clarified this by revising the caption of Fig.S14 as follows:

“Fig.S14. The colonization outcome prediction of *E. faecium* and *A. muciniphila* in human stool-derived *in vitro* microbial communities using the community's relative species richness as the only predictor. Here, the relative species richness, denoted as r , of a microbial community is computed as the ratio between the number of species present in this community and that present in all baseline communities. For each of the 6-fold cross-validations (ROC curves shown in a light color), we simply used $(1 - r)$ as the

colonization probability to compute its AUROC (or Accuracy). For colonization outcomes of *E. faecium*, the average AUROC (or Accuracy) based on the relative species richness is 0.78 (or 0.32), respectively. By contrast, the average AUROC (or Accuracy) of the Random Forest classifier based on the taxonomic profile is 0.86 (or 0.82), respectively (see **Fig. 3G**). For colonization outcomes of *A. muciniphila*, the average AUROC (or Accuracy) based on the relative species richness is 0.74 (or 0.53), respectively. By contrast, the average AUROC (or Accuracy) of the Random Forest classifier based on the taxonomic profile is 0.84 (or 0.78), respectively (see **Fig. 4G**).”

Note that, due to the imbalanced colonization outcomes for *E. faecium* (and *A. muciniphila*), Accuracy is not an ideal metric to quantify the prediction performance.

Point 3.4: In addition, the description of Fig. S14 analysis in the main text (lines 249-250) leaves out the crucial quantitative information that it is only *slightly* worse than the more elaborate abundance-based models (a difference in AUC of <0.1, and equal or better than logistic regression results in Figs. 3 and 4). I think it would be helpful to list the AUROC (+ accuracy) of this diversity-based model in the main text, since it is critical for evaluating the advantage of the more models based on species abundance.

Response: We thank Reviewer #3 for this excellent suggestion. We have revised that part of the manuscript as follows:

“For comparison, we used the community diversity (quantified by the relative species richness) as the only feature to predict colonization outcome (see **Fig.S14**). Our results indicated that the relative species richness alone can be used as a predictor, but its prediction performance (with average AUROC=0.78 and Accuracy=0.32) is worse than elaborate classifiers, e.g., Random Forest using the taxonomic profile (with average AUROC=0.86 and Accuracy=0.82). Overall, our colonization experiments of *E. faecium* in complex human gut microbial communities validated that the data-driven approach can solve the classification problem of COP.”

Point 3.5: In their response to point #13 from my last review, the authors added a new citation to the Bunin (2017) paper. However, the citation seems to have been added in a somewhat strange place (line 132, describing the setup of the GLV simulations) rather than the introduction of Eq. 4 (where it is most relevant). This omission, coupled with the “discovered” language used in the text, might give the misleading impression that Eq. 4 is a novel result.

Response: We thank Reviewer #3 for this comment. We have cited the Bunin (2017) paper right after the introduction of Eq.4.

Point 3.6: Likewise, in their response to point #17 from my previous review, the authors added citations to the Buffie et al 2015 and Venturelli et al 2018 papers in the introduction, but again in a rather oblique way (line 54) in a separate discussion that is distinct from the discussion of data driven approaches on line 80 where they are most relevant. Without acknowledging this prior work, I worry that the paragraph on lines 72-80 does not accurately situate the present study within the current state of the field. I think it is important to include some discussion of these existing approaches in the Introduction around line 80, even if it means that the Intro needs to be expanded somewhat (it is already quite short, so I think there should be plenty of space).

Response: We thank Reviewer #3 for this constructive comment. We have added the discussion of these references in a more relevant place (see main text, lines 79-83):

“While statistical models have been used to decipher microbial interactions in synthetic human gut microbial communities³⁶ and mouse gut microbiota³⁰, the use of data-driven models has not received much attention in microbial ecology^{37, 38}.”

Minor:

Point 3.7: I thank the authors for including the details about how the antibiotics were added to the experiments (point #7 in my previous review). The description in the response letter is great, but the edits to the text could be modified slightly to indicate the length of passaging that was performed before colonization started. Also, it would be helpful to repeat or elaborate on this description in the methods section (e.g. around line 522) for readers who might overlook it within the main text.

Response: We thank Reviewer #3 for this constructive comment. We have added this detail in the main text and the Methods section.

“Those stool-derived communities were treated with antibiotics for 24h on Day 0 (i.e. a single pulse). Afterwards, the communities were passaged five times (from Day 1 to Day 6) in antibiotic-free medium to reach a stable state before introducing the exogenous species.”

Point 3.8: Line 94 claims that “Our results suggest that the function of complex microbial communities can be predicted via data-driven approaches and tunable.” I don’t really see how this claim is supported by the present study (since it doesn’t directly look at community function or tunability). Please tone down or clarify.

Response: We thank Reviewer #3 for this insightful comment. We have revised the description accordingly:

“Our results suggest that the colonization outcome of complex microbial communities can be predicted via data-driven approaches and tunable.”

Point 3.9: On line 85, it would be helpful to clarify that this is $O(N)$ colonization outcomes *per colonizing species* (otherwise, it runs the risk of getting confused with the $O(N)$ vs $O(N^2)$ issue).

Response: We thank Reviewer #3 for this insightful comment. We have revised the description accordingly.

“Nevertheless, with a sample size on the order of $\sim O(N)$ per colonizing species, machine learning models were able to achieve accurate classification of colonization outcomes in synthetic data (AUROC > 0.8).”

Reviewer #2 (Remarks on code availability):

Point 2.0: I only did a cursory scan of the github repository, but it looks like all of the key elements are there and the README is file is highly detailed. I did not try to install or run the code.

Response: We thank Reviewer #2 for reviewing our manuscript again. We are glad to know that the reviewer has been satisfied with our previous response.